



Atmospheric
Measurement
Techniques

# Validation of SMILES HCl profiles over a wide range from the stratosphere to the lower thermosphere

**Seidai Nara**[1,2]**, Tomohiro O. Sato**[1]**, Takayoshi Yamada**[1]**, Tamaki Fujinawa**[3]**, Kota Kuribayashi**[4]**, Takeshi Manabe**[5]**, Lucien Froidevaux**[6]**, Nathaniel J. Livesey**[6]**, Kaley A. Walker**[7]**, Jian Xu**[8]**, Franz Schreier**[8]**, Yvan J. Orsolini**[9]**, Varavut Limpasuvan**[10]**, Nario Kuno**[2,11]**, and Yasuko Kasai**[1,2]

[1]National Institute of Information and Communications Technology, Nukui-kita, Koganei, Tokyo, Japan
[2]University of Tsukuba, Tennodai, Tsukuba, Ibaraki, Japan
[3]National Institute for Environmental Studies, Onogawa, Tsukuba, Ibaraki, Japan
[4]Mynavi Corporation, Tokyo, Japan
[5]Osaka Prefecture University, Gakuen-cho, Naka-ku, Sakai, Osaka, Japan
[6]Jet Propulsion Laboratory, California Institute of Technology, Pasadena, CA, USA
[7]Department of Physics, University of Toronto, 60 St. George Street, Toronto, Ontario, Canada
[8]German Aerospace Center (DLR), Remote Sensing Technology Institute, Oberpfaffenhofen, Germany
[9]Norwegian Institute for Air Research (NILU), Kjeller, Norway
[10]School of the Coastal Environment, Coastal Carolina University, Conway, SC, USA
[11]Tomonaga Center for the History of the Universe, University of Tsukuba, 1–1–1 Tennodai, Tsukuba, Ibaraki, Japan

**Correspondence:** Y. Kasai (ykasai@nict.go.jp)

**Abstract.** Hydrogen chloride (HCl) is the most abundant (more than 95 %) among inorganic chlorine compounds $Cl_y$ in the upper stratosphere. The HCl molecule is observed to obtain long-term quantitative estimations of the total budget of the stratospheric chlorine compounds. In this study, we provided HCl vertical profiles at altitudes of 16–100 km using the Superconducting Submillimeter-Wave Limb-Emission Sounder (SMILES) from space. The HCl vertical profile from the upper troposphere to the lower thermosphere is reported for the first time from SMILES observations; the data quality is quantified by comparison with other measurements and via theoretical error analysis. We used the SMILES level-2 research product version 3.0.0. The period of the SMILES HCl observation was from 12 October 2009 to 21 April 2010, and the latitude coverage was 40° S–65° N. The average HCl vertical profile showed an increase with altitude up to the stratopause ($\sim$ 45 km), approximately constant values between the stratopause and the upper mesosphere ($\sim$ 80 km), and a decrease from the mesopause to the lower thermosphere ($\sim$ 100 km). This behavior was observed in all latitude regions and reproduced by the Whole Atmosphere Community Climate Model in the specified dynamics configuration (SD-WACCM). We compared the SMILES HCl vertical profiles in the stratosphere and lower mesosphere with HCl profiles from Microwave Limb Sounder (MLS) on the Aura satellite, as well as from the Atmospheric Chemistry Experiment Fourier Transform Spectrometer (ACE-FTS) on SCISAT and the TErahertz and submillimeter LImb Sounder (TELIS) (balloon borne). The TELIS observations were performed using the superconductive limb emission technique, as used by SMILES. The globally averaged vertical HCl profiles of SMILES agreed well with those of MLS and ACE-FTS within 0.25 and 0.2 ppbv between 20 and 40 km (within 10 % between 30 and 40 km; there is a larger discrepancy below 30 km), respectively. The SMILES HCl concentration was smaller than those of MLS and ACE-FTS as the altitude increased from 40 km, and the difference was approximately 0.4–0.5 ppbv (12 %–15 %) at 50–60 km. The difference between SMILES and TELIS HCl observations was about 0.3 ppbv in the polar winter region between 20 and 34 km, except near 26 km. SMILES HCl error sources that may cause discrepancies with the other ob-

servations are investigated by a theoretical error analysis. We calculated errors caused by the uncertainties of spectroscopic parameters, instrument functions, and atmospheric temperature profiles. The Jacobian for the temperature explains the negative bias of the SMILES HCl concentrations at 50–60 km.

## 1 Introduction

Hydrogen chloride (HCl) is the most abundant species of all total inorganic chlorine compounds $Cl_y$ in the stratosphere. More than 95 % of $Cl_y$ exists as HCl at about 1 hPa ($\sim$ 50 km) (see Froidevaux et al., 2008). Thus, HCl has been used for long-term quantitative estimates of the total budget of stratospheric chlorine (WMO, 2010). Stratospheric HCl vertical profiles have been observed globally by several satellites. The Halogen Occultation Experiment (HALOE) on the Upper Atmosphere Research Satellite measured stratospheric HCl profiles continuously from late 1991 to November 2005 (Russell III et al., 1996). Comparisons of HCl profiles obtained by the Aura Microwave Limb Sounder (MLS) and the Atmospheric Chemistry Experiment Fourier Transform Spectrometer (ACE-FTS) have been reported by Froidevaux et al. (2008) and Mahieu et al. (2008). These profiles agreed within 5 % (0.15 ppbv) at 0.5 hPa ($\sim$ 53 km). The Aura/MLS and ACE-FTS HCl concentrations were larger (by 10 %– 20 %) than those from HALOE (WMO, 2010). Balloon-borne remote sensing in the polar winter region was performed using the TErahertz and submillimeter LImb Sounder (TELIS) from 2009 to 2011 (Birk et al., 2010). Xu et al. (2018) reported the error analysis and comparison of some trace gas profiles provided by TELIS.

The Superconducting Submillimeter-Wave Limb-Emission Sounder (SMILES) was launched in 2009 to observe the atmospheric compositions of ozone and species related to the stratospheric ozone destruction cycle, including HCl (Kikuchi et al., 2010). The SMILES mission is a joint project of the Japan Aerospace Exploration Agency and the National Institute of Information and Communications Technology (NICT). SMILES is an instrument used to conduct limb observations from the International Space Station (ISS) using supersensitive 4 K superconducting receivers in the submillimeter-wave regions (625 and 649 GHz bands). The non-Sun-synchronous circular orbit of the ISS gave us an opportunity for observations of HCl profiles at different local times. The unprecedented low noise of the SMILES instrument provided a sensitivity 10 times superior to those of previous microwave/submillimeter limb emission instruments used to observe HCl spectra from space (Kikuchi et al., 2010). The SMILES observations were used to reveal small abundances of atmospheric species in the stratosphere and mesosphere (e.g., Sato et al., 2017; Yamada et al., 2020).

A comparison of the HCl profiles inside the Antarctic vortex has been performed for 19–24 November 2009 using the SMILES, Aura/MLS, and ACE-FTS data (Sugita et al., 2013). The SMILES HCl values (version 2.1.5 of NICT products) agreed within 10 % (0.3 ppbv) with those of the MLS and ACE-FTS HCl data between the potential-temperature levels of 450 and 575 K, and 425 and 575 K, respectively. In contrast to the previous study, this study used 3.0.0 (v300) SMILES data and provided a global comparison, including the stratosphere and mesosphere. The HCl climatology using SMILES NICT version 2.1.5 was reported by Kreyling et al. (2013).

In this study, we examined HCl vertical profiles from 16 to 100 km, including the upper mesosphere and lower thermosphere (MLT) region. We used the SMILES NICT level-2 product v300, which was released in late 2012 (http://smiles.nict.go.jp/pub/data/index.html, last access: 17 October 2020). HCl vertical profiles from the upper troposphere to the lower thermosphere are reported for the first time. We perform a validation of the SMILES HCl vertical profiles by comparisons with the corresponding global model results of SD-WACCM, satellite observations from MLS and ACE-FTS, and balloon-borne observations from TELIS, and we provide a SMILES HCl error analysis.

The SMILES HCl observations and retrieval procedure are described in Sect. 2. The HCl vertical profile derived by the SMILES product and comparisons of the HCl distribution between 40° S and 60° N with SD-WACCM are shown in Sect. 3. Results of HCl comparisons between SMILES and other instruments are described in Sect. 4. An estimation of the systematic and random errors is presented in Sect. 5. Section 6 describes our conclusions.

## 2 SMILES HCl observations

The SMILES instrument had been attached to the Japanese Experimental Module (JEM) on the ISS since September 2009. SMILES operational period started on 12 October 2009 and ended on 21 April 2010. The ISS has a non-Sun-synchronous circular orbit with an inclination angle of 51.6° with respect to the Equator. The height of the ISS changed slowly over the observational period ranging from 340 to 360 km. We tilted the line of sight at a 45° angle in the direction of the forward movement of the ISS to observe the northern polar region. The SMILES parameters are summarized in Table 1. The details of the SMILES observations are described in Kikuchi et al. (2010) and in an ozone-validation paper by Kasai et al. (2013), respectively.

Herein, we briefly describe the SMILES atmospheric observations in terms of the HCl spectra. SMILES had three frequency bands in the submillimeter-wave regions: band A (624.32–625.52 GHz), band B (625.12–626.32 GHz), and band C (649.12–650.32 GHz). Each single scan provided a combination of two out of three of the frequency bands:

**Table 1.** SMILES specification

| Parameter | Characteristics |
| --- | --- |
| Orbit | Non-Sun-synchronous orbit |
|  | Inclination angle 51.6° |
|  | Altitude 340–360 km |
|  | ~ 90 min orbital period |
| Latitude coverage | 38° S–65° N (nominal) |
| Measurement geometry | Limb scan |
| Number of scans | 1630 scans per day |
| Integration time | 0.47 s |
| Frequency range | 624.32–625.52 GHz (band A) |
|  | 625.12–626.32 GHz (band B) |
|  | 649.12–650.32 GHz (band C) |
| Receiver system | SIS[a] mixer and HEMT[b] amplifiers |
| Spectrometer | Acousto-optical spectrometers |
|  | (AOS1 and AOS2) |
| Frequency resolution | 0.8 MHz |
| System noise temperature | ~ 350 K |

[a] Superconductor–insulator–superconductor. [b] High-electron-mobility transistor.

bands A + B, C + B, or C + A. The combinations changed on a daily basis, as described in Kikuchi et al. (2010). The rotational transitions of two HCl isotopologues were located in bands B and A for $H^{35}Cl$ at 625.9 GHz and $H^{37}Cl$ at 624.9 GHz, respectively. Two acousto-optical spectrometers (AOSs) were equipped in the SMILES instrument. The spectral resolution obtained by AOS1 and AOS2 was 0.8 MHz. The band configuration for each HCl isotopologues had three patterns as follows:

– $H^{37}Cl$ observation by band A with AOS1,

– $H^{37}Cl$ observation by band A with AOS2, and

– $H^{35}Cl$ observation by band B with AOS2.

Band B was always associated with AOS2.

Figure 1 shows the number of observations per day obtained for $H^{35}Cl$ (band B) and $H^{37}Cl$ (band A) for 5° latitude bins during the SMILES observational period. It should be noted that the sampling is not homogeneously distributed for the SMILES observations. The SMILES instrument was sometimes not in operation, for example, when the ISS was boosted up from a low to a high altitude, and the solar panels disturbed the observational line of sight. The SMILES observational latitude range was 65° N–38° S in nominal operations and changed to 38° N–65° S when the ISS performed a yaw maneuver of 180°, which happened three times during the 7-month observational period.

We used the SMILES NICT level-2 product v300 for this study. The general method of v300 for the HCl retrieval is similar to that of the level-2 product version 2.1.5 (v215) (Kasai et al., 2013). The major updates from the v215 product are as follows: (1) an improvement in the spectrum calibration, particularly for the gain non-linearity of the receiver

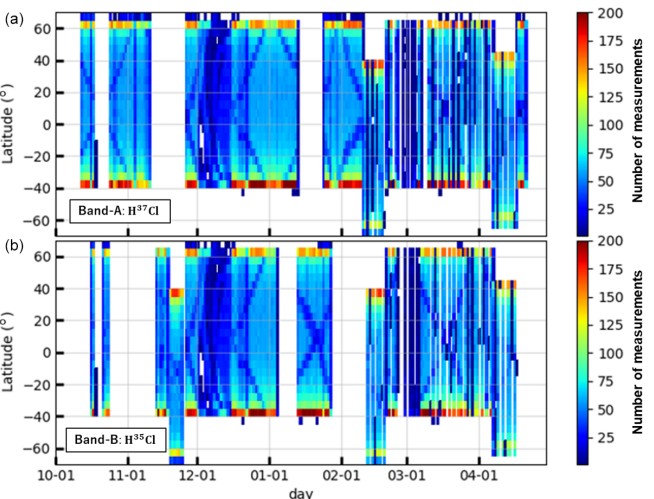

**Figure 1.** Number of HCl observations for a 5° latitude bin in 1 d for the range of SMILES observational latitudes and period (12 October 2009–21 April 2010). Panels **(a)** and **(b)** show the number of observations from band A ($H^{37}Cl$) and band B ($H^{35}Cl$), respectively.

system (Ochiai et al., 2013), (2) an improvement in the accuracy of the tangent height estimation (Ochiai et al., 2013), and (3) an update of the temperature retrieval. The temperature profile used in the SMILES v300 retrieval was synthesized, assuming the hydrostatic equilibrium, using the Goddard Earth Observing System version 5 (GEOS-5) reanalysis meteorological datasets and the climatology based on the Aura/MLS measurements (Kuribayashi et al., 2017). The GEOS-5 datasets were used in the upper troposphere and stratosphere, and the Aura/MLS datasets were used in the mesosphere and lower thermosphere, respectively. The temperature profile was also retrieved using the ozone transition in the SMILES v300 retrieval process, but it was not applied to the retrieval of atmospheric species including HCl, except for ozone, to avoid systematic error propagation issues (SMILES-NICT, 2014). Level-1b spectrum data version 008 were used for the SMILES NICT level-2 product v300 (Ochiai et al., 2013). The spectroscopic parameters for the HCl retrieval were based on Cazzoli and Puzzarini (2004) and Baron et al. (2011), as summarized in Table 2. We used only the $H^{35}Cl$ data because the intensities of the $H^{37}Cl$ spectra were weaker than those of $H^{35}Cl$, as shown in Fig. 2.

The criteria for the selection of the $H^{35}Cl$ profiles used in this study are as follows:

1. $\chi^2$ less than 0.8. $\chi^2$ is defined by

$$\chi^2 = \left[ y - F(x) \right]^T S_\epsilon^{-1} \left[ y - F(x) \right]$$
$$+ \left[ x - x_a \right]^T S_a^{-1} \left[ x - x_a \right]. \qquad (1)$$

$y$ is the observed spectrum, $S_\epsilon$ is the covariance matrix of the measurement noise, $x_a$ is the a priori state,

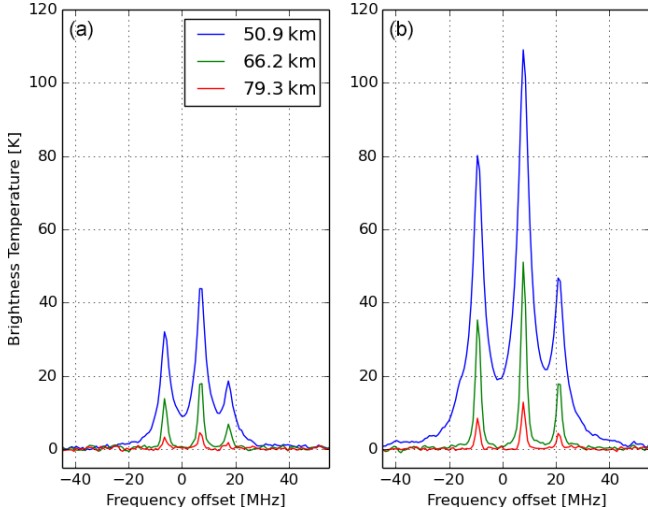

**Figure 2. (a)** An example of the $H^{37}Cl$ spectra observed by SMILES from a single-scan measurement. Tangent heights are approximately 50, 65, and 80 km. Center frequency is 624.98 GHz. **(b)** An example of the $H^{35}Cl$ spectra for the same condition as for the $H^{37}Cl$ spectra. Center frequency is 625.92 GHz.

**Table 2.** Spectroscopic line parameters measured in the laboratory and used in the processing.

| Species | Line frequency (MHz) | Air-broadening (MHz/Torr)/ temperature dependence | Frequency shift (MHz) |
|---|---|---|---|
| $H^{35}Cl$ | 625 901.6627[a] 625 918.7020[a] 625 931.9977[a] | 3.39/0.72[b] | 0.145[b] |
| $H^{37}Cl$ | 624 964.3718[a] 624 977.8059[a] 624 988.2727[a] | 3.39/0.72[b] | 0.145[b] |

[a] Laboratory measurements by Cazzoli and Puzzarini (2004). [b] Baron et al. (2011).

$S_a$ is the covariance matrix of $x_a$, and $F(x)$ is the forward model depending on the state vector $x$. We used the US standard atmosphere profiles as the a priori state ($x_a$) (US-Standard, 1976). They are used separately for polar, equatorial, summer midlatitude, or winter midlatitude regions. The forward model is essentially the atmospheric radiative transfer and the instrument model.

2. Measurement response (MR) between 0.8 and 1.2 for each vertical grid. MR is defined by

$$MR[i] = \sum_j \mathbf{A}[i, j]. \tag{2}$$

$\mathbf{A}$ is the averaging kernel matrix. A value of MR near unity indicates that most of the information in the retrieval results is provided by observations. A low value

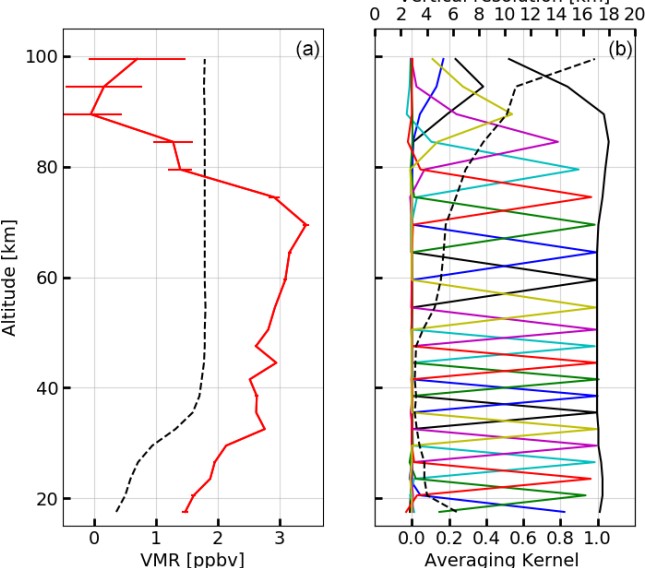

**Figure 3. (a)** An example of a HCl profile retrieved from the $H^{35}Cl$ spectrum on 15 November 2009 at a latitude of 38.3° N, longitude of 12.7° E, and SZA of 160.9°. The solid red line indicates the retrieved HCl volume mixing ratio (VMR), and the error bar is the root sum square of the smoothing and measurement errors. The dashed black line is an a priori HCl profile used for the retrieval procedure. Panel **(b)** shows the averaging kernel, the measurement response (indicated by the thick black line), and the FWHM of the averaging kernel profile (indicated by the thick dashed black line).

of MR indicates that the retrieval results are largely influenced by the a priori state and are forced to be identical to a priori values.

An example $H^{35}Cl$ profile and its averaging kernel are presented in Fig. 3 for the single-scan spectrum observation at latitude of 38.3° N, longitude of 12.7° E, and solar zenith angle (SZA) of 160.9° on 15 November 2009. We assumed that the natural isotopic abundance of $H^{35}Cl$/HCl was 0.7576 by Berglund and Wieser (2011) for these retrievals. The peak values of the averaging kernels for the SMILES $H^{35}Cl$ profiles exceeded 0.8 in the height range of 16 to 90 km ($\sim$ 100 to 0.001 hPa) as shown in Fig. 3. The full width at half maximum (FWHM) of the averaging kernels is approximately 3–4 km at altitudes of 18–50 km and becomes greater than 5 km for the altitudes 50–90 km.

## 3 Vertical and latitudinal distribution of SMILES HCl

In this section, we provide the HCl vertical profiles derived from the SMILES NICT level-2 product v300. Figure 4a shows the HCl zonal mean distribution in the latitudinal range from 40° S to 60° N. The time period is between 16 October 2009 and 17 April 2010. The HCl vertical distribution shows an increase with altitude with a maxi-

mum below the stratopause ($\sim 45$ km), approximately constant values between the stratopause and the upper mesosphere ($\sim 80$ km), and a decrease with altitude from the mesopause to the lower thermosphere ($\sim 100$ km). In the lower and middle stratosphere, HCl is generated by the reaction of Cl with $CH_4$ and $HO_2$ and transported by circulation (e.g., Brewer–Dobson circulation). The HCl abundance is balanced by production ($Cl + HO_2 \rightarrow HCl + O_2$) and loss ($HCl + OH \rightarrow Cl + H_2O$, $HCl + h\nu \rightarrow H + Cl$) in the upper stratosphere and mesosphere. Near the mesopause, the photodissociation becomes the dominant reaction, and the HCl abundance decreases with height (Brasseur and Solomon, 2005).

This behavior was reproduced by the Whole Atmosphere Community Climate Model version 4 (WACCM4) in the specified dynamics configuration (SD-WACCM) (Lamarque et al., 2012; Limpasuvan et al., 2016); see Fig. 4b. In the specified dynamics configuration, the simulated meteorological fields in the troposphere and stratosphere are constrained to the Global Modeling and Assimilation Office Modern-Era Retrospective Analysis for Research and Applications (Rienecker et al., 2011). The horizontal resolution of SD-WACCM simulation is 1.9° and 2.5° for latitude and longitude, respectively. The vertical grid is from the ground to 150 km with 88 levels, and the time resolution is 3 h. Differences of HCl abundance between SMILES and SD-WACCM are shown in Fig. 4c. For these comparisons, the SD-WACCM profiles are interpolated linearly for each SMILES observation point and time from the nearest model positions and times. Figure 4d shows the number of selected SMILES observations. The SMILES HCl abundance agrees well with the SD-WACCM simulation (within 0.1 ppbv) in the stratosphere and middle mesosphere. At 70–100 km, the difference of the HCl VMR between SMILES and WACCM in the upper mesosphere and lower thermosphere is 0.4–1.0 ppbv, which is larger than the total systematic error in the SMILES HCl VMR estimated by quantitative error analysis (0.1–0.4 ppbv). Thus, the difference is significant. It might be caused by a combination of the SMILES observation and the WACCM model uncertainties. The underestimation of the model could be because of uncertainties in the HCl photodissociation and the reaction with OH radical, which are dominant destruction mechanisms of HCl for the mesosphere/thermosphere (Brasseur and Solomon, 2005).

## 4 Comparison of SMILES HCl profile with those obtained using other instruments

We performed a comparison between SMILES HCl products and other datasets obtained using the MLS on the Aura satellite, the ACE-FTS on the SCISAT satellite, and the balloon-borne TELIS instrument. The characteristics of instruments and datasets are summarized in Table 3. The mean absolute difference, $\Delta_{abs}$, for each altitude between SMILES and the

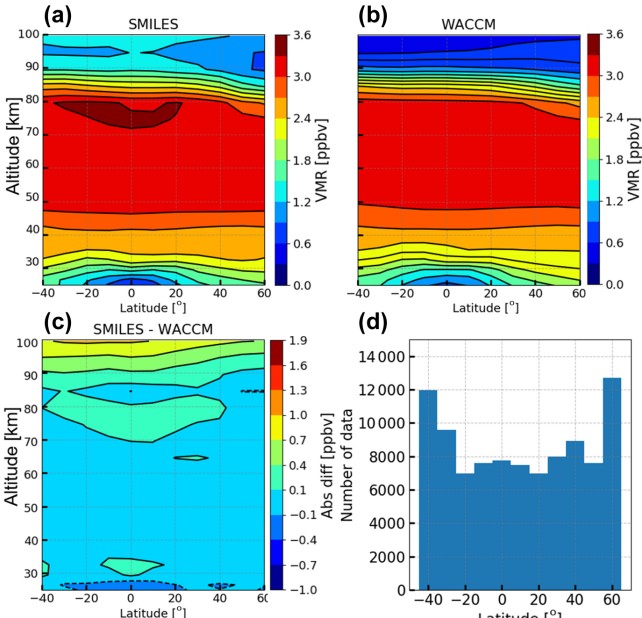

**Figure 4.** Zonal mean distribution of HCl from SMILES, SD-WACCM, and their differences (SMILES – SD-WACCM) are shown in panels **(a)**, **(b)**, and **(c)**, respectively. Each panel displays distributions as a function of latitude ($x$ axis) and altitude ($y$ axis). The latitude range covers between 40° S and 60° N. The number of HCl profiles observed by SMILES is described in panel **(d)** as a histogram versus latitude. The time period is between 16 October 2009 and 17 April 2010. The data are averaged within latitude bins of 10°.

other instrument is defined as

$$\Delta_{abs}(z) = \frac{1}{N(z)} \sum_{i=1}^{N(z)} (x_{c,i}(z) - x_{s,i}(z)), \tag{3}$$

where $x_{s,i}(z)$ and $x_{c,i}(z)$ are the HCl volume mixing ratio (VMR) of $i$th coincidence at an altitude $z$ for SMILES and the other instrument, respectively. $N(z)$ is the number of coincidence at an altitude $z$. The mean relative difference in percentage, $\Delta_{rel}$, for each altitude is given by

$$\Delta_{rel}(z) = \frac{1}{N(z)} \sum_{i=1}^{N(z)} \frac{x_{c,i}(z) - x_{s,i}(z)}{(x_{c,i}(z) + x_{s,i}(z))/2} \times 100. \tag{4}$$

The coincidence criteria between the SMILES observations and those of the other instruments were set to within 2° for latitude, 8° for longitude, and 5 h for time, following the previous work on MLS observations (Froidevaux et al., 2008).

### 4.1 Comparison with Aura/MLS

The Aura satellite has a Sun-synchronous orbit with an ascending node at 13:45 LT (Waters et al., 2006). The measurements cover latitudes from 82° S to 82° N and provide

**Table 3.** Characteristics of instruments and datasets used in the comparison.

| Instrument and platform | Measurement type – | Time period overlapped with SMILES | Data version | Number of coincidence | Altitude range | Vertical resolution |
|---|---|---|---|---|---|---|
| MLS Aura | Limb emission Sub-mm/microwave | 24 Jan 2010 –27 Jan 2010 (band 13) | 4.2 | 4356 | 100–0.32 hPa | 3–5 km |
| ACE-FTS SCISAT | Solar occultation Mid-IR | 16 Oct 2009 –17 Apr 2010 | 4.0 | 935 | 7–63 (Equator) 6–59 km (polar) | 3–4 km |
| TELIS Balloon borne | Limb emission Sub-mm/THz | 24 Jan 2010 | 3v02 | 4 | 16–34 km | 1.8–3.5 km |

approximately 3500 vertical profiles each day. We used version 4.2 of the Aura/MLS HCl profiles for these comparisons. The Aura/MLS team provided two HCl products: one from spectral band 13 (HCl-640-B13) and one from band 14 (HCl-640-B14). The HCl-640-B13 product is available for the pressure region between 100 and 0.32 hPa. The single profile precision of HCl-640-B13 is less than 0.8 ppbv (25 %) in the stratosphere, and the estimated accuracy is approximately 0.3 ppbv (10 %) (Livesey et al., 2018).

The vertical and horizontal resolutions are 3–5 km and 200–400 km, respectively. The vertical resolution of MLS is of the same order as that of SMILES. We used the recommended parameters "status," "quality," and "convergence" for screening the Aura/MLS data based on Livesey et al. (2018). The good profiles were selected using (1) quality < 1.2 TS1, (2) convergence > 1.05 TS2, and (3) status is even. The SMILES profiles were linearly interpolated onto the MLS pressure grid for the comparisons.

Figure 5 shows a comparison of the HCl vertical profiles derived from SMILES and Aura/MLS in the SMILES latitude range. The total number of coincident profile pairs was 4356. The HCl profiles from the SMILES measurements agreed with those of MLS within 0.25 ppbv between 20 and 40 km (50–3 hPa), while large biases of about 0.4 ppbv were confirmed between 40 and 55 km (3–0.32 hPa). The differences between HCl profiles increase from the stratosphere to the mesosphere. Figure 6 shows the zonal mean absolute differences averaged for 5° latitude bins. The differences did not change with latitude in the altitude region between 35 and 55 km (7–0.32 hPa), while below 35 km altitude, several areas of large differences can be identified, especially in the equatorial region.

Figure 7 displays more detailed features of the differences between the SMILES and MLS profiles. Figure 7a shows the averaged HCl vertical profiles of SMILES (blue) and MLS (red), and the absolute difference in each latitude region (40–20° S, 20° S–20° N, 20–50° N, and 50–65° N). Figure 7b shows the latitudinal distribution averaged in 5° bins for SMILES (blue) and MLS (red) at 30, 40, and 50 km. The error bar shows the 1σ variability for each latitude grid. In the equatorial regions, the SMILES HCl profiles were 0.5 ppbv

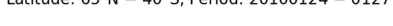

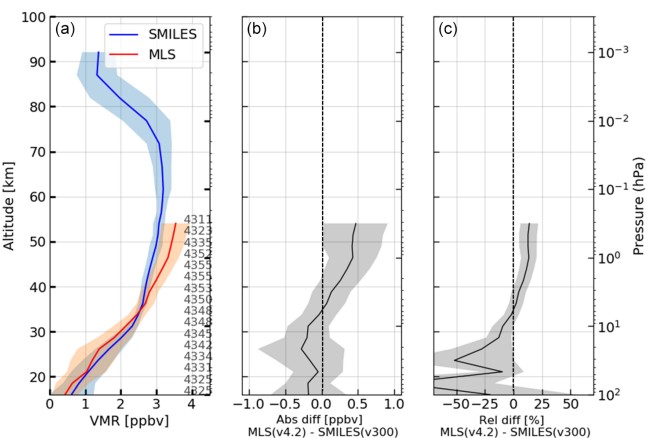

**Figure 5.** Comparison between SMILES and MLS profiles from 24 to 27 January 2010. **(a)** Mean HCl VMR values (solid lines) and 1σ (shaded areas) for SMILES and MLS. The blue and red lines indicate the SMILES HCl profiles and MLS profiles, respectively. **(b)** The absolute difference between the SMILES and MLS profiles calculated using Eq (3). **(c)** The relative difference between the SMILES and MLS profiles calculated using Eq (4).

larger than those of MLS at 30 km (10 hPa). On the other hand, the HCl profiles from the SMILES measurements were 0.4 ppbv lower than those of MLS in all latitude regions at 50 km.

## 4.2 Comparison with SCISAT/ACE-FTS

The ACE-FTS is an instrument mounted on the Canadian SCISAT satellite. SCISAT moves along an orbit at a 650 km altitude and is inclined at 74° to the Equator (Bernath et al., 2005). The ACE comprises two instruments: the Fourier Transform Spectrometer (ACE-FTS) and the Measurement of Aerosol Extinction in the Stratosphere and Troposphere Retrieved by Occultation (ACE-MAESTRO). The HCl observation has been performed using ACE-FTS. The ACE-FTS observes solar occultation spectra in the infrared spectral region (750–4400 cm$^{-1}$) with a high spectral resolution (0.02 cm$^{-1}$). We used ACE-FTS HCl profiles from the ver-

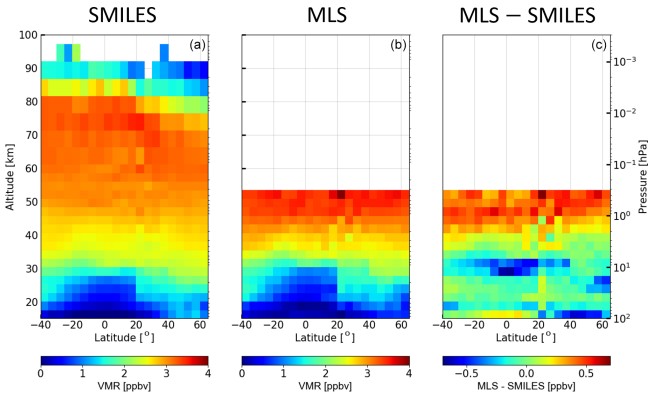

**Figure 6. (a)** SMILES zonal mean HCl profiles versus latitude. Profiles that satisfied coincidence criteria between SMILES and MLS for 24–27 January 2010 were used here. **(b)** Zonal mean HCl profiles from MLS. **(c)** The absolute difference of the matched pairs versus latitude.

sion 4.0 software, which is the latest data version (Boone et al., 2020). The vertical resolution of the ACE-FTS HCl retrieval is 3–4 km, and the values from the retrieval grid are interpolated onto the 1 km grid, using a piecewise quadratic method (Bernath et al., 2005). The vertical resolution of ACE-FTS is of the same order as that of SMILES. In this study, we used the data within $\pm 3$ times the median absolute deviation (3-MAD) from the median to remove significantly large positive and negative biases based on Boone et al. (2019).

Figure 8 presents the comparison of the HCl vertical profiles between SMILES and ACE-FTS in the latitude range of 40° S–65° N. The total number of coincident data pairs was 935. The SMILES HCl profiles agreed with those of the ACE-FTS by 0.2 ppbv between 20 and 40 km (50–2 hPa), while 0.5 ppbv (15 %) lower than those of ACE-FTS above 40 km. This negative bias in the SMILES HCl concentration was confirmed as in the case of the MLS–SMILES comparison. The results of the comparison with the ACE-FTS for each latitudinal region are presented in Fig. 9a in a manner similar to Fig. 7a. In the tropical region, a large discrepancy was observed at an altitude of 25 km because of relativity poor sampling number. The HCl profiles observed using the SMILES were about 0.5 ppbv (15 %) less than those of ACE-FTS for each latitudinal region above 50 km.

The period of the SMILES $H^{35}Cl$ observation (from 16 October 2009 to 17 April 2010) was covered by the ACE-FTS observation period, while the MLS band-13 observation period had an overlap of only 4 d. We analyzed the difference between SMILES and ACE-FTS for each month and confirmed the seasonal variation of the bias. Figure 9b shows a seasonal variation of the difference between the SMILES and ACE-FTS profiles in the Northern Hemisphere (30–65° N latitude range). The upper panels in Fig. 9b show a mean of the SMILES (blue) and ACE-FTS (green) profiles for

each month at three altitude levels. The difference for every month, shown in the lower panels, was represented by the mean for each month at each altitude, and the error bar showed the $1\sigma$ standard deviation. No significant seasonal dependence of the difference was observed. The difference of HCl value from these measurements was consequently about 0.5 ppbv at 50 km.

### 4.3 Comparison with balloon-borne instrument TELIS

TELIS (Birk et al., 2010) is a balloon-borne THz/submillimeter-wave spectrometer with superconductive 4 K technology similar to that of SMILES. TELIS was one of the instruments used in the balloon observation campaigns for three winters (2009–2011) in Kiruna, Sweden. Limb observations were performed from a flight altitude of 30–35 km, toward the stratosphere and the upper troposphere with 1.5–2 km vertical sampling. TELIS was used to observe $H^{37}Cl$ at the transition frequency of 1873.4 GHz in the 1.8 THz channel (level-1b version 3v02), with corrected non-linearity problems in the radiometric calibration using a quadratic term for each frequency segment. The details on the TELIS 1.8 THz channel and its L1 data processing are shown in Suttiwong (2010).

In this study, we compared two $H^{37}Cl$ profiles obtained by TELIS (observation number 20044 and 21537) with the SMILES profiles (band B and identifiers 761 and 762) in terms of geolocation over Kiruna (67.8° N, 20.4° E) and time on 24 January 2010. The total error of the $H^{37}Cl$ profile derived from the TELIS measurement was estimated to be 0.25–0.5 ppbv, with a vertical resolution of 3 km (Xu et al., 2018).

Figure 10 shows a comparison of the HCl vertical profiles of SMILES and TELIS. Figure 10a–c show the SMILES and TELIS HCl vertical profiles, their absolute difference, and their relative difference, respectively. We converted the $H^{37}Cl$ amount to HCl using the natural isotopic abundance ($H^{37}Cl / HCl = 0.2424$). The SMILES and TELIS HCl profiles agreed well, within 0.5 ppbv from 17 to 34 km. A depletion in the HCl profile was observed below 25 km in both the SMILES and TELIS observations. This result is considered to have been caused by chlorine activation in the polar vortex (Webster et al., 1993; Wegner et al., 2016). Both SMILES and TELIS profiles agree well in general, but the TELIS profiles are larger than the SMILES profiles above 32 km (8 hPa). The $H^{37}Cl$ line is still rather strong at higher altitudes, and the dominant error source of the TELIS data stems from the non-linearity in the calibration process, which shows that even a small uncertainty may result in significant errors in the retrieval (Xu et al., 2018). Table 4 summarizes the results of the comparison with other observations.

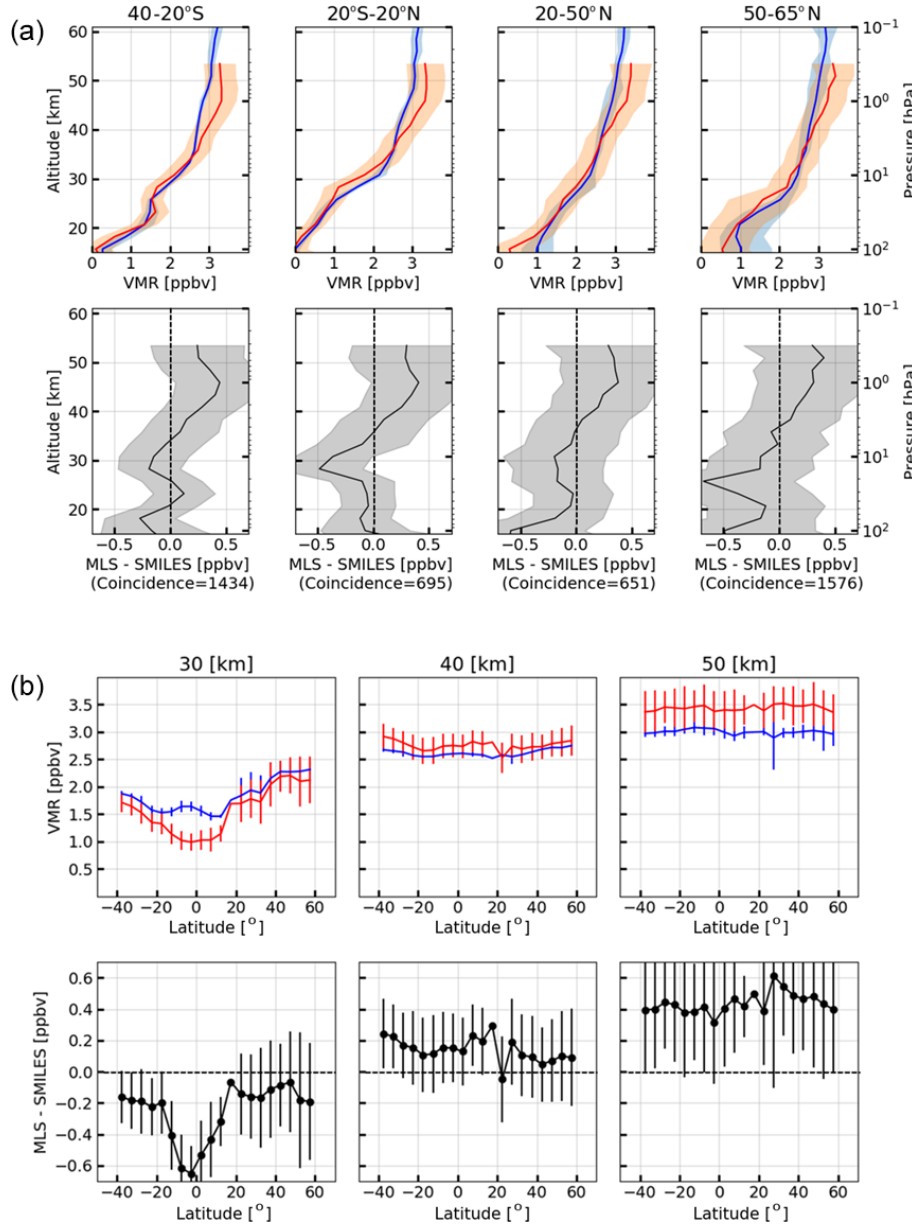

**Figure 7. (a)** Mean HCl profiles, $1\sigma$ uncertainty based on variability (shaded region) and the absolute differences for each latitude region. Upper row: the blue and red lines indicate the mean profiles from SMILES and MLS for 40–20° S, 20° S–20° N, 20–50° N, and 50–65° N (left to right panels). Lower row: the absolute difference between the SMILES and MLS profiles calculated using Eq. (3) for each latitude region. **(b)** Latitudinal variation of SMILES and Aura/MLS HCl profiles at three altitude levels. The profiles were averaged for each 5° bin. Upper row: mean of SMILES (blue) and MLS (red) profiles for each altitude level. The error bars indicate $1\sigma$ uncertainties. Lower row: the means and $1\sigma$ of the absolute differences are displayed.

## 5 Theoretical error analysis

We have evaluated the total error in the HCl vertical profiles observed by SMILES, and we discussed the cause of the bias observed in the comparison study; see Sect. 4.

## 5.1 Estimation of total error

We employed a perturbation method to estimate the total error of the SMILES HCl profile. The details of the perturbation method of the SMILES error analysis have been described in Kasai et al. (2006) and Sato et al. (2014), Kasai et al. (2013), and Sato et al. (2012) for ozone isotopes, ozone, and ClO, respectively. We assumed an averaged HCl profile

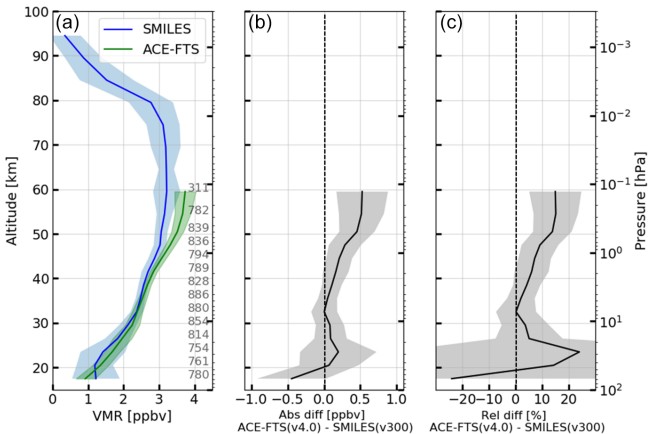

**Figure 8.** Comparison between SMILES and ACE-FTS profiles from 16 October 2009 to 17 April 2010. **(a)** The mean HCl VMR values (solid lines) and $1\sigma$ (shaded areas) for SMILES and ACE-FTS. The blue and green lines are the SMILES and ACE-FTS HCl profiles, respectively. **(b)** The absolute difference between the SMILES and ACE-FTS profiles calculated using Eq. (3). **(c)** The relative difference between the SMILES and ACE-FTS profiles calculated using Eq. (4).

**Table 4.** Summary of the HCl comparison study between SMILES and the other instruments.

| Altitude (km) | $\Delta x_{\text{MLS}}$ (ppbv) | $\Delta x_{\text{ACE-FTS}}$ (ppbv) | $\Delta x_{\text{TELIS}}$ (ppbv) |
|---|---|---|---|
| 60 | – | 0.5 | – |
| 50 | 0.4 | 0.4 | – |
| 40 | 0.2 | 0.15 | – |
| 30 | −0.2 | 0.05 | 0.4 |
| 20 | −0.1 | −0.02 | 0.5 |

within the coincidence with MLS in the southern midlatitude region (20–40° S) as a reference. The total error ($E_{\text{total}}$) for each altitude grid was calculated by

$$E_{\text{total}}[i] = \sqrt{E_{\text{noise}}[i]^2 + E_{\text{smooth}}[i]^2 + E_{\text{param}}[i]^2}, \quad (5)$$

where $E_{\text{noise}}$ is the error due to spectrum noise, $E_{\text{smooth}}$ is the smoothing error, and $E_{\text{param}}$ is the model parameter error. $E_{\text{noise}}$ and $E_{\text{smooth}}$ were calculated by

$$E_{\text{noise}}[i] = \sqrt{\mathbf{S}_{\text{noise}}[i,i]} \quad (6)$$

$$\mathbf{S}_{\text{noise}} = \mathbf{D}\mathbf{S}_\epsilon\mathbf{D}^{\mathrm{T}} \quad (7)$$

and

$$E_{\text{smooth}}[i] = \sqrt{\mathbf{S}_{\text{smooth}}[i,i]} \quad (8)$$

$$\mathbf{S}_{\text{smooth}} = (\mathbf{U} - \mathbf{A})\mathbf{S}_{\text{a}}(\mathbf{U} - \mathbf{A})^{\mathrm{T}}, \quad (9)$$

where $\mathbf{D}$ is the contribution function, and $\mathbf{U}$ is the unit matrix.

We calculated the errors due to the uncertainties of the spectroscopic parameters and the instrument functions to calculate $E_{\text{param}}$. The error $E_{\text{param}}$ was calculated by

$$E_{\text{param}} = I\left(y_{\text{ref}}, b_0 + \Delta b_0\right) - I\left(y_{\text{ref}}, b_0\right) \quad (10)$$

where $I$ is the inversion function, $b_0$ is the vector of model parameters, and $\Delta b_0$ is the uncertainty on model parameters. $y_{\text{ref}}$ is the reference spectrum calculated using the reference profile. The details of the estimation of the total error are described in Sato et al. (2012). The error sources and perturbations for the model parameter used in this study are summarized in Table 5. These parameters and perturbation values are based on Sato et al. (2014). The uncertainties of the spectroscopic parameters of the $O_3$ transition 625.37 GHz were included to estimate the error due to the interference from the line shape of $O_3$ spectrum. The calibration error was not considered in this study because the latest L1b data (version 008) were used, and Sato et al. (2014) reported that the error due to the spectrum calibration in these L1b data was insignificant.

The estimated errors in the SMILES HCl v300 product are presented in Fig. 11. The total model parameter error, labeled as "param", was calculated by a root sum square (RSS) of all model parameter errors. The three spectroscopic parameters, line intensity, air-pressure-broadening coefficient ($\gamma_{\text{air}}$), and its temperature dependence ($n_{\text{air}}$) were dominant error sources below 30 km. The $\gamma_{\text{air}}$ was a major error source between 30 and 60 km, which was about 0.09 ppbv ($\sim 3\,\%$) at 50 km. At altitudes of 60–90 km, the largest error source was the AOS response function, and its peak value reached 0.38 ppbv ($\sim 12\,\%$) at 70 km.

### 5.2 Discussion: cause of the negative bias of the SMILES HCl vertical profile

In this section, we discuss the cause of the negative bias of the SMILES HCl vertical profile, i.e., approximately 10 % less than those of Aura/MLS and ACE-FTS especially at the altitudes between 40 and 60 km. Such a large bias could not be explained by the total error estimated by the perturbation method described in Sect. 5.1. We further investigated the cause of this bias by looking at the temperature profile used in the retrieval calculation. The temperature profile used for the retrieval procedure of SMILES was lower than those of MLS and ACE-FTS particularly in the upper stratosphere and mesosphere. We estimated the difference of the retrieved HCl vertical profile, $\Delta x$, due to the difference of the temperature profile, $\Delta T$, as follows:

$$\mathbf{\Delta}x = \mathbf{D}\mathbf{K}_T\mathbf{\Delta}T, \mathbf{D} = \frac{\partial x}{\partial y}, \mathbf{K}_T = \frac{\partial y}{\partial T}. \quad (11)$$

The Jacobian $\mathbf{K}_T$ indicates the sensitivity of the spectral brightness temperature ($y$) with reference to changes of the temperature ($T$). Here, we synthesized the Jacobian with a perturbation of 0.5 K. Figure 12a shows the Jacobian as a

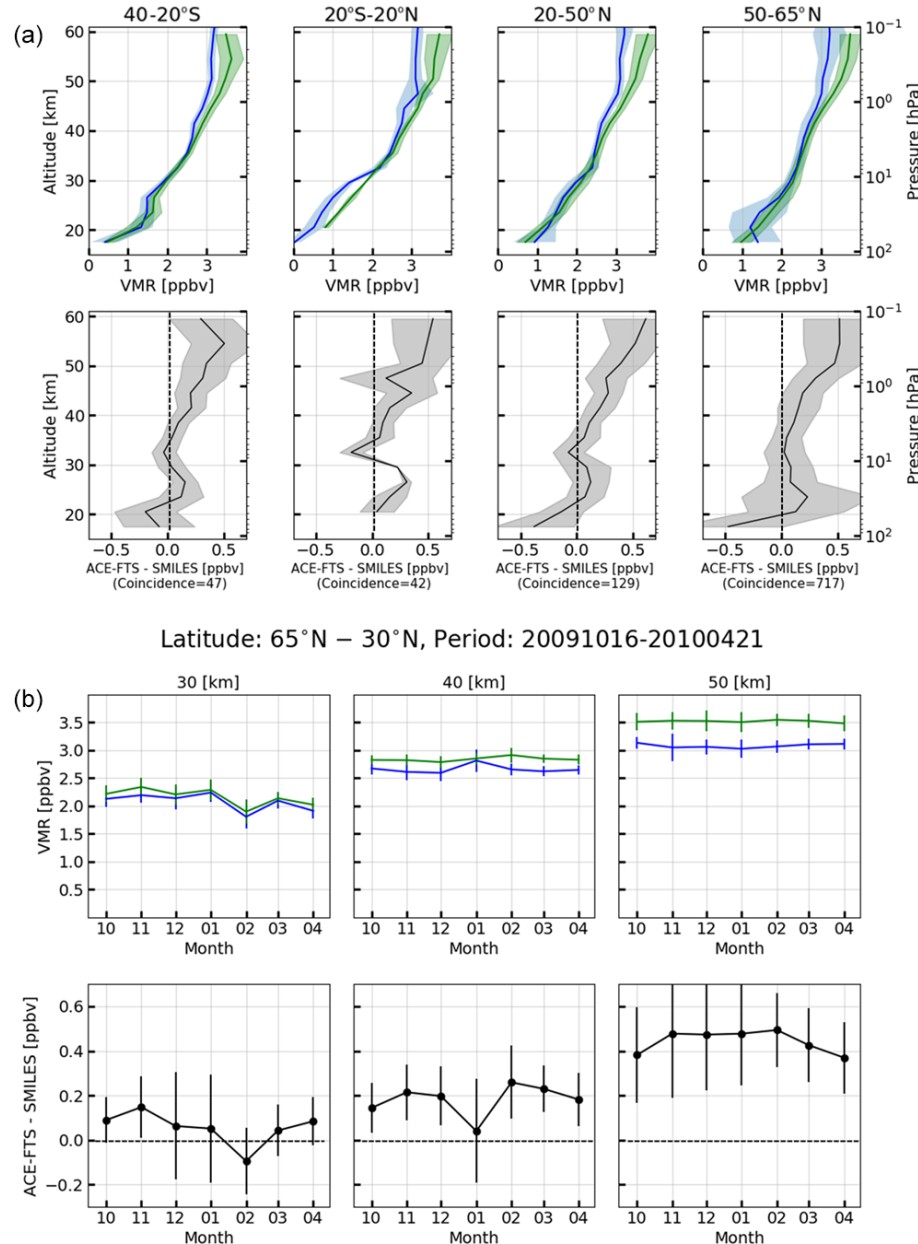

**Figure 9. (a)** The mean of HCl values, $1\sigma$, and the absolute difference for each latitudinal region are shown. Upper row: the blue and green lines indicate the SMILES HCl profiles and ACE-FTS profiles for each latitudinal region (40–20° S, 20–20° N, 20–50° N, and 50–65° N) from the left side. Lower row: the absolute difference between the SMILES and ACE-FTS profiles calculated using Eq. (3) for each latitudinal region. **(b)** Seasonal variation in the SMILES and ACE-FTS HCl profiles at three altitude levels. Latitudinal range is 30–65° N. Upper row: monthly mean of SMILES (blue) and ACE-FTS (green) HCl profiles for each altitude level. The error bars correspond to $1\sigma$. Lower row: the mean and $1\sigma$ uncertainty of the absolute difference.

function of tangent height. Here, the minimum value of the column of the $\mathbf{K}_T$ matrix is plotted. A negative Jacobian value means that higher temperatures induce a lower brightness temperature spectrum, thus increasing the HCl abundance in the retrieval to compensate for this underestimation. The temperature profiles used in the retrievals by SMILES, MLS, and ACE-FTS and their differences (MLS−SMILES,

ACE-FTS−SMILES) are shown in Fig. 12b and c, respectively. The vertical profile of temperature of SMILES is approximately 5–10 K lower than those of both MLS and ACE-FTS for altitudes between 50 and 60 km. The a priori temperature profile used in the SMILES retrieval procedure is based on the GEOS-5 profile in the stratosphere and MLS retrieved profile in the mesosphere and above. The altitude

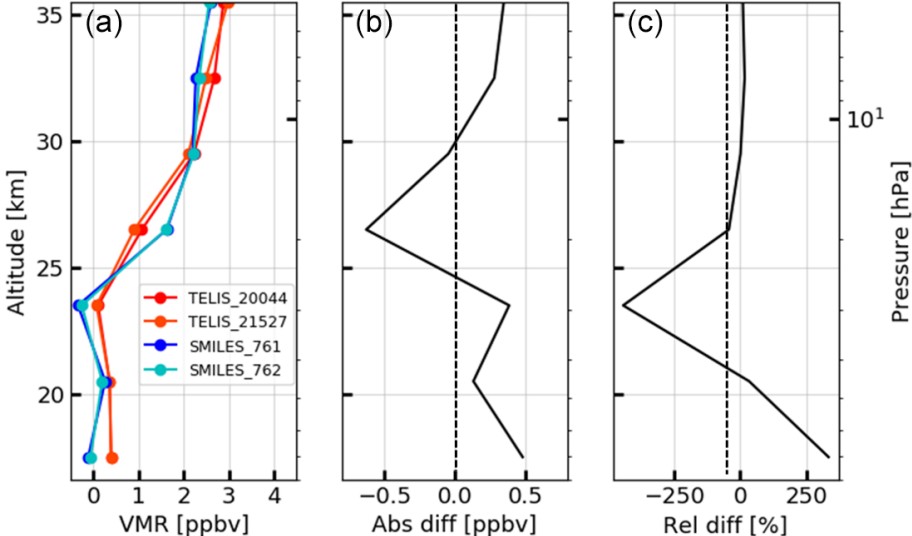

**Figure 10.** Differences between SMILES and TELIS HCl observations are shown for 24 January 2010. Two of the SMILES observations were taken from closer points, where the latitude, longitude, and SZA were 64.3° N, 30.6° E, and 84.6° SZA and 64.8° N, 38.2° E, 86.2° SZA, respectively. **(a)** Each profile obtained by SMILES and TELIS instruments; **(b)** the absolute difference between the SMILES and TELIS profiles calculated using Eq. (3); **(c)** the relative difference between the SMILES and TELIS profiles calculated using Eq. (4).

**Table 5.** Error sources and their perturbation values for this study.

| Error source | Uncertainty |
| --- | --- |
| Spectroscopic parameters of the $H^{35}Cl$ transition at 625.92 GHz | |
| Line intensity | 1 % |
| Air-pressure-broadening coefficient ($\gamma_{air}$) | 3 % |
| Temperature dependence of $\gamma_{air}$ ($n_{air}$) | 10 % |
| Spectroscopic parameters of the $O_3$ transition at 625.37 GHz | |
| $\gamma_{air}$ of the $O_3$ transition ($\gamma_{air}(O_3)$) | 3 % |
| $n_{air}$ of the $O_3$ transition ($n_{air}(O_3)$) | 10 % |
| Instruments | |
| Antenna beam pattern (antenna) | 2 % |
| AOS response function (AOS) | 5 % |

The perturbation value was based on Sato et al. (2014).

limit of the MLS temperature profile is 0.001 hPa, with a vertical resolution of 6–14 km and a precision of 1.2–3.6 K per profile. MLS used GEOS-5 up to 1 hPa as with SMILES. For pressures smaller than 1 hPa, the Committee on Space Research (COSPAR) International Reference Atmosphere (CIRA-86) is used as a priori temperature information (with a loose constraint) in the MLS retrieval procedure (Schwartz et al., 2008). The altitude range and vertical resolution of the CIRA-86 profile are ground to 120 and 2 km, respectively (Fleming et al., 1990). The MLS temperature profiles in the stratosphere and above were retrieved primarily from bands near the 118 GHz $O_2$ spectral line (Livesey et al., 2018). The temperature value retrieved by MLS is 10 K lower than the a priori profile on average in some areas for pressure values smaller than 1 hPa, based on earlier version validation studies (Schwartz et al., 2008). The ACE-FTS HCl retrieval procedure uses the temperature profile retrieved from the ACE-FTS measurements between 18 and 125 km. The ACE-FTS temperature profiles were retrieved from $CO_2$ VMR using hydrostatic equilibrium (Boone et al., 2020). The vertical resolution of ACE-FTS retrieved temperature is 3–4 km. The temperature values retrieved by ACE-FTS are less than 10 K larger than the MLS derived temperatures (Schwartz et al., 2008). These types of differences are also seen in the comparison results performed here.

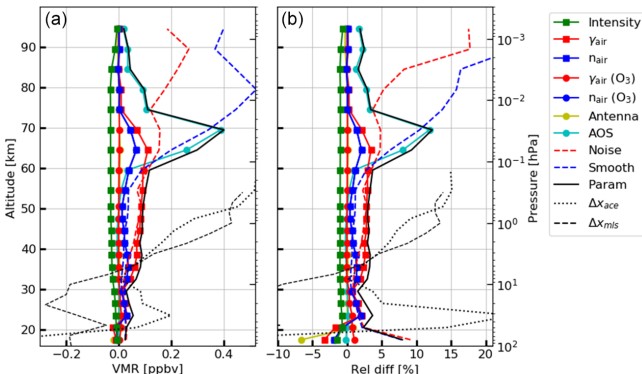

**Figure 11.** Summary of the errors for a single-scan observation. Panels **(a)** and **(b)**, respectively, show the absolute difference and relative difference, which were estimated using the perturbation method. The green marker indicates an error due to the uncertainty in line intensity ("intensity"). The red and blue squares show errors for the air-broadening coefficient ("$\gamma_{air}$") and its temperature dependence ($n_{air}$) of the H$^{35}$Cl transition. The red and blue circles show errors for "$\gamma_{air}$" and $n_{air}$ of the O$_3$ transition ("$\gamma_{air}$ (O$_3$)", "$n_{air}$ (O$_3$)"). The yellow and cyan symbols show errors for the antenna beam pattern ("antenna"), and the AOS response function ("AOS"). The dashed red and blue lines indicate $E_{noise}$ ("noise") and $E_{smooth}$ ("smooth"), respectively. The solid black lines ("param") indicate the total model parameter errors, obtained from the RSS of all the estimated parameter errors. The dotted and dashed black lines indicate the difference between SMILES and MLS, and between SMILES and ACE-FTS.

Figure 12d shows the $\Delta x$ due to the difference of the temperature profile. This temperature difference caused an increase in SMILES HCl of 0.12 and 0.20 ppbv at 50–60 km for the MLS and ACE-FTS comparisons, respectively. In the comparison study (see Sect. 4), we confirmed the negative bias in the SMILES HCl vertical profile of $0.4 \pm 0.38$ and $0.5 \pm 0.28$ ppbv between 40 and 60 km for MLS and ACE-FTS, respectively. We estimated the total error by the perturbation method and investigated the temperature profile used in the retrieval calculation, in order to investigate the cause of this negative bias. The largest error sources were the uncertainties in $\gamma_{air}$ of the H$^{35}$Cl transition and the temperature profiles used in the retrievals. We assumed a 3 % uncertainty in $\gamma_{air}$ which could lead to a $-0.1$ ppbv bias in the 40–60 km region. In addition to the error due to the $\gamma_{air}$ coefficients, the effect of temperature differences should be taken into account at altitudes above 50 km. The gradual increase in bias is caused by the difference in altitude at which these two errors become more pronounced. The difference in temperature profiles used in the retrieval between SMILES and ACE-FTS caused a negative bias of about 0.2 ppbv at 50–60 km. In summary, 0.3 in 0.5 ppbv (60 %) negative bias between SMILES and ACE-FTS can be explained by the uncertainty in $\gamma_{air}$ and the temperature profiles used in the SMILES retrievals. The effect of $\gamma_{air}$ error on the negative bias between SMILES and MLS is less than that of SMILES

and ACE-FTS, since SMILES and MLS observed the same H$^{35}$Cl transition lines and the values of the $\gamma_{air}$ are consistent within approximately 1 % (3.39 MHz/Torr for SMILES and 3.42 MHz/Torr for MLS; Drouin, 2004). A 1 % difference in $\gamma_{air}$ might cause the HCl abundance to increase by about 0.03 ppbv according to our error analysis with a perturbation method. About 40 % (0.15 out of 0.4 ppbv) of the negative bias between SMILES and MLS can be explained. Therefore, our theoretical error analysis shows that the SMILES HCl has a negative bias of at most 0.25 ppbv between 40 and 60 km; the remaining difference between SMILES and MLS or ACE-FTS can be explained by the standard deviation in the comparison result.

## 6 Conclusions

In this study, the HCl vertical profile in a wide range from the upper troposphere to the lower thermosphere was reported for the first time using the SMILES NICT level-2 data product v300. The HCl distribution shows an increase with altitude with a maximum below the stratopause ($\sim 45$ km), approximately constant values between the stratopause and the upper mesosphere ($\sim 80$ km), and a decrease with altitude from the mesopause to the lower thermosphere ($\sim 100$ km). In the lower and middle stratosphere, HCl is generated by the reaction of Cl with CH$_4$ and HO$_2$ and transported by circulation (e.g., Brewer–Dobson circulation). The HCl abundance is balanced by production (Cl + CH$_4$ $-$ > HCl + CH$_3$, Cl + HO$_2$ $\rightarrow$ HCl + O$_2$) [TS3] and loss (HCl + OH $\rightarrow$ Cl + H$_2$O, HCl + h$\nu$ $\rightarrow$ H + Cl) in the upper stratosphere and the mesosphere. Above the mesopause, the photodissociation becomes the dominant reaction and the HCl abundance decreases. This behavior was reproduced by the SD-WACCM model, and the SMILES HCl vertical profile agreed well with the SD-WACCM model within $\pm 0.1$ ppbv for altitudes between 30 and 70 km.

The data quality of the SMILES HCl vertical profile was quantified by comparisons versus other measurements and supported by a theoretical error analysis. We compared the SMILES HCl vertical profiles with well-validated data of two satellite instruments (Aura/MLS and ACE-FTS), as well as a balloon-borne instrument (TELIS) at their temporal–spatial coincidences. The SMILES HCl profiles at 20–40 km showed good agreement, within less than $0.25 \pm 0.3$ $(1\sigma)$ and $0.20 \pm 0.2$ $(1\sigma)$ ppbv versus MLS and ACE-FTS, respectively. The comparison with TELIS in the polar winter region at 20–34 km showed similar behavior with differences within 0.3 ppbv, which is the same order of magnitude as the systematic error of the TELIS data. A negative bias ($< 0.5$ ppbv) of the SMILES HCl profiles from 40 to 60 km altitudes was observed in comparisons versus MLS and ACE-FTS HCl profiles.

We estimated the total error for SMILES HCl based on the perturbation method and considering the uncertainties in

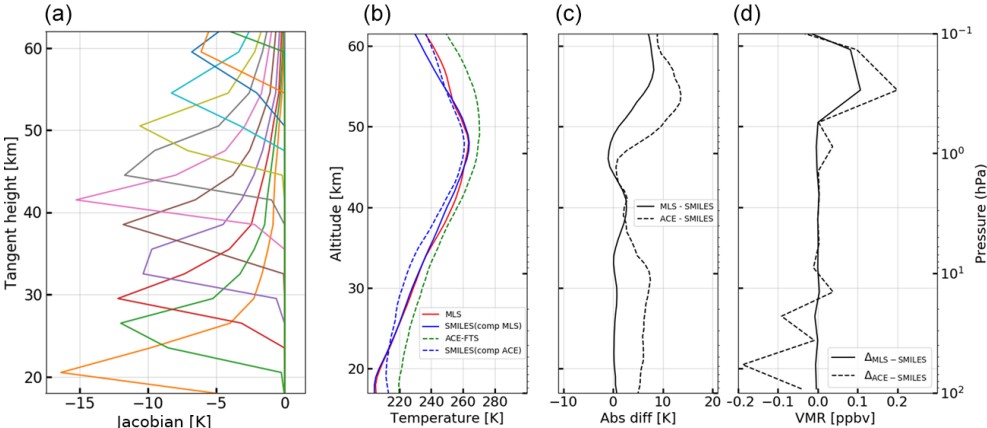

**Figure 12. (a)** The lines represent the Jacobian for the temperature calculated using the mean profile from the SMILES winter midlatitude region (20–40° S). The units of the Jacobian (*x* axis) are Kelvin. **(b)** Temperature profile of each instruments. The solid blue and red lines indicate the mean SMILES and MLS temperature profiles within the coincidence criteria used for the HCl comparisons. The dashed blue and green lines indicate the SMILES and ACE-FTS temperature profiles for the same coincidence criteria. **(c)** The absolute difference of temperature profiles between SMILES and other instruments calculated using Eq. (3). **(d)** The difference in HCl profiles calculated using Eq. (11).

atmospheric temperature profiles used in the retrievals. The dominant contributions to the systematic errors were from the air-broadening parameter (0.09 ppbv) and the AOS response function (0.38 ppbv) at 30–60 km and 60–100 km altitudes, respectively. The uncertainty in the temperature profile used in the retrieval calculation caused a negative bias of 0.12 to 0.20 ppbv between 50 and 60 km, which was 30 % and 40 % of the HCl abundance difference between SMILES and MLS, and SMILES and ACE-FTS, respectively. The uncertainties of the air-broadening parameter and the temperature profile are capable of contributing a total of 40 %–50 % of the SMILES HCl negative biases at 50–60 km. In summary, our theoretical error analysis showed that the HCl profiles had a negative bias of 0.20–0.25 ppbv at 50–60 km, which is consistent with the observed differences versus MLS and ACE-FTS profiles within 1 standard deviation. The spectroscopic parameters of the HCl transitions and the temperature profile above the stratopause are key parameters for potential improvements in the SMILES retrieval algorithms.

The observation of HCl abundances in the upper atmosphere is important to investigate the long-term total budget of anthropogenic chlorine in the Earth's atmosphere. Further observations and model studies are needed to better understand the sources and sinks, transport processes, and chemical reactions related to HCl.

*Data availability.* The SMILES data are available at http://smiles. nict.go.jp/pub/data/index.html (last access: 18 November 2020; SMILES-NICT, 2020). The MLS data are available at https: //disc.gsfc.nasa.gov/datasets?page=1&keywords=AURAMLS (last access: 18 November 2020; NASA, 2020) or see https://mls.jpl. nasa.gov/data/ (last access: 18 November 2020; Aura/MLS, 2020).

The ACE-FTS data are available at http://www.ace.uwaterloo.ca/ data.php (last access: 18 November 2020; SCISAT/ACE-FTS, 2020). The details of the TELIS 1.8 THz channel and its L1 data processing are shown at https://elib.dlr.de/66749/ (last access: 18 November 2020; Suttiwong, 2010) "Development and characterization of the balloon borne instrument TELIS (TErahertz and submillimeter LImb Sounder): 1.8 THz receiver Suttiwong, Nopporn (2010)" and/or https://elib.dlr.de/97249/ (last access: 18 November 2020; Xu, 2015) "Inversion for Limb Infrared Atmospheric Sounding Xu, Jian (2015)". The WACCM data are available at http:// www.cesm.ucar.edu/working_groups/Whole-Atmosphere/ (last access: 18 November 2020; WACCM, 2020).

*Author contributions.* SN designed the study and performed the analysis. YK designed the study and provided the SMILES data. TOS provided the code for the error analysis and contributed to data analysis and interpretation. TY, TF, and KK contributed to the data analysis and reviewed the manuscript. LF, NJL, KAW, JX, and FS provided MLS, ACE-FTS, and TELIS information and knowledge of HCl in the atmosphere and helped review the manuscript. YJO and VL performed and provided the WACCM simulations, and helped review the manuscript. NK and TM supervised writing of and reviewed the manuscript.

*Competing interests.* The authors declare that they have no conflict of interest.

*Special issue statement.* This article is part of the special issue "New developments in atmospheric Limb measurements: Instruments, Methods and science applications (AMT/ACP inter-journal SI)". It is a result of the 10th international limb workshop, Greifswald, Germany, 4–7 June 2019.

*Acknowledgements.* The JEM/SMILES mission is a joint project of the Japan Aerospace Exploration Agency and the National Institute of Information and Communications Technology. The authors wish to acknowledge the contributions made by their colleagues at JAXA and NICT for managing and supporting the SMILES mission. The authors also thank Kazuya Muranaga (Systems Engineering Consultants Co., Ltd.) and Jan Moller (Molflow Co., Ltd.) for supporting the data processing in the SMILES NICT level-2 product. Work at the Jet Propulsion Laboratory, California Institute of Technology, was performed under a contract with the National Aeronautics and Space Administration. The Atmospheric Chemistry Experiment (ACE), also known as SCISAT, is a Canadian-led mission mainly supported by the Canadian Space Agency.

*Financial support.* This research has been supported by the Ministry of Internal Affairs and Communications (grant nos. 0155-0285 and 0155-0093). Yasuko Kasai is supported by the Funding Program for Next Generation World-Leading Researchers (NEXT Program) (grant no. GR101).

*Review statement.* This paper was edited by Miriam Sinnhuber and reviewed by two anonymous referees.

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

**Remarks from the typesetter**