# Peer review of "Validation of the vertical profiles of HCl over the wide range of the stratosphere to the lower thermosphere measured by SMILES"

_Atmospheric Measurement Techniques, 2020_

## Referee Comment (RC1) · Anonymous Referee #2 · 15 Jun 2020

Summary: This paper presents new data of HCl vertical profiles obtained by the SMILES instrument operational between October 2009 and April 2010 on the International Space Station. The HCl vertical profiles cover an extended range from the upper troposphere/lower stratosphere up to the upper mesosphere/lower thermosphere region. The retrieval technique is discussed and results are presented and compared to output from the WACCM model. The main part of the paper consists of a validation study that features comparisons of the SMILES HCl with measurements from the MLS instrument on the Aura Satellite, the ACE-FTS on the Scisat-1 satellite, and the TELIS balloon-borne instrument. Overall agreement is good although significant differences are found that are most pronounced in the upper stratosphere/lower mesosphere re-

gion. The paper suggests that differences in the temperature profile used for trace gas retrieval from the thermal emission measurements are responsible for a significant part of the differences between SMILES and the other satellite measurements.

Review: This is an interesting paper that fits well into the scope of Atmospheric Measurement Techniques. The most interesting part of the presented material is the vertical range over which HCl profiles can be retrieved due to the high spectral resolution and signal-to-noise ratio achieved by the measurement technique. The coverage in the mesosphere and in the transition zone to the lower thermosphere makes this dataset unique and I can envision significant interest in it for example for studies of the dynamics in this atmospheric region, which is otherwise difficult to access. My main point of criticism of this paper is the lack of a description of the temperature retrieval that underlies the trace gas retrieval. This is particularly important as the authors identify differences in the underlying temperature profile as the main reason for differences in HCl in the upper stratosphere/lower mesosphere region. A description of the temperature data, including the method of the temperature retrieval, its vertical range and its accuracy in the various altitude regions is essential for understanding possible caveats of the HCl data, especially for use in studies of the mesosphere and lower thermosphere, where other data is sparsely available. With the addition of this information, as well as responses to my more detailed comments below, I would recommend this paper for publication.

Detailed comments:

Abstract, last sentence: The last sentence of the abstract should be moved further up, somewhere in the first third of the abstract. Acronyms should be introduced also in the abstract.

Line 86: This is the location where a description of the temperature retrieval should be provided. It will set the stage for a lot of the discussion further below. It should answer questions such as: How is the temperature retrieved? What is the vertical range of the

temperature retrieval? How accurate is it, especially in the mesosphere?

Line 110: "an increasing with the altitude increased" should be "an increase with altitude"

Line 111: "decreased with the altitude increasing" should be "decreased with altitude"

Line 116 concerning differences between SMILES and WACCM in the mesosphere: Is there an explanation for these differences? How does the high HCl near the mesopause compare with Cly at lower altitudes? Is this realistic? A depiction of the HCl/Cly ratio based SMILES HCl and Cly estimates from lower altitudes in comparison to the WACCM HCl/Cly ratio may be interesting.

Lines 138-139: How can the accuracy be smaller than the precision? In my understanding the precision is largely due to statistical errors, while the accuracy also includes systematic errors and hence should be larger than the precision. This should be corrected or explained.

Lines 155-156: The discussion of water vapor as a source for differences is not convincing. There is very little water vapor in the stratosphere around 30 km.

Line 166: What is "MAD"? Abbreviations should be introduced.

Figure 5: For the right panel an x-axis range at which relative difference does not go off scale should be used.

Line 176: "overlapped" -> "overlap"

Line 177: "conformed" -> "confirmed"

Line 199: The reference to Webster is ok for the mechanism but is there also a reference to the development of chlorine activation for that particular winter. That would be more relevant.

Line 201-202: Is there a reference that supports this statement about calibration uncertainties and their impact on the TELIS retrieval?

Figure 10: Same comment as for figure 5 - for the right panel an x-axis range at which relative difference does not go off scale should be used.

Table 4: I don't understand the statement in the footnote. The comparison didn't actually look that bad at these altitudes.

Line 242: The paper states that "The temperature profile used for the retrieval procedure of SMILES was smaller than those of MLS and ACE-FTS particularly in the upper stratosphere, mesosphere, and lower thermosphere", however, no temperatures for upper mesosphere, lower thermosphere are shown. This should be added, probably in figure 12. It should be stated how the temperatures for the other satellite measurements were derived. Instead of the word "smaller", I'd maybe suggest "lower" or "colder".

Line 253: "at the altitude (of) 50–70km." – Again, no data is shown for 70 km altitude.

Figure 12: The altitude scale is different in panel a compared to panels b, c, d. These panels should be plotted over the same vertical range avoid confusion. Also, the figure quality is inferior, with the lines and text kind of blurry even when zoomed in, so the figure should be redone.

Lines 260-265, discussion of bias at 55 km: What about other altitudes? Figs. 5 and 8 suggest that differences start around 40 km and increase gradually with altitude. However, Fig. 12 suggests that the temperatures around 40 km are very consistent. This behavior should be included in the discussion.

Line 292: "Totally," – I suggest maybe "In Summary,"

Line 295: "above stratosphere" – I suggest "above the stratopause"

---

## Referee Comment (RC2) · Anonymous Referee #1 · 17 Jul 2020

This paper presents a new hydrogen chloride (HCl) data set, derived from SMILES measurements performed from October 2009 to April 2010. HCl plays a key role in the stratospheric ozone destruction mechanisms and observations of this species are essential to quantify the total budget of stratospheric chlorine. The vertically resolved data presented in this paper covers the altitude range 16-100 km, from the lower stratosphere to the lower thermosphere. This is the first time that HCl is observed in such a broad vertical range, making this new data set particularly valuable. The SMILES HCl retrieval procedure is described, and the resulting profiles are compared to independent observational data sets as well as to model simulations. Those comparisons are discussed, in light of the error analysis included in the study. I recommend the

publication of this paper in AMT, after consideration of the minor revisions suggested below.

(For your information, I have not read Referee #2's report before writing mine, for the sake of integrity. There might therefore be some redundant comments.)

General comments

The paper presents an important new data set for middle atmospheric studies. The retrieval process is presented clearly, and the validation method is valid. The overall presentation is well structured. However, some important explanation is missing. For example, you say that the water vapour effect is a possible cause for the observed latitudinal differences, but you do not explain anything about this effect, or you describe the HCl vertical and geographical distribution, without discussing the chemical and dynamical mechanisms controlling it. Last but not least, you conclude that the temperature data used to retrieve HCl is responsible for up to 30 to 40% of the observed differences between SMILES and the other instruments, but you do not give any information about the source and quality of these data sets. Please see my specific comments below for more detailed questions and suggestions. Moreover, the language should be improved. See my suggestions for technical corrections below.

Specific comments

p.1 l.12-14: Please specify also the relative values corresponding to the given absolute differences.

p.2 l.43-46: Could you please explicitly comment the differences between the results of the comparisons mentioned here (between SMILES v2 and MLS / ACE-FTS, by Sugita et al., 2013) and the results of your validation study? It would be interesting to add such a comment in the conclusion section, where you discuss your results.

p.4. l.76: Do not forget to specify in the text that you are talking about the daily number of observations.

p.4 l.92: Could you give some information about the a priori data used in the retrieval process: what is the source of this data set? Does it depend on latitude and time? Etc.

p.6 l.102: "We assumed that the natural isotopic abundance of H35Cl/HCl was 0.7576...": Please explain where this value comes from. A citation should be added here.

p.6-7 Sect.3: Please discuss the differences and/or similarities between the three sub-periods under consideration or change Fig. 4 to show only the results averaged over the whole SMILES operational period. I do not understand the point in dividing the comparisons between SMILES and SD-WACCM into three different time periods if this is not discussed.

p.6 l.110-112: An explanation about the HCl vertical and latitudinal distribution is missing. Please describe the chemical and physical mechanisms controlling it, or at least comment on the current state of knowledge about that. (This could be added either here or in the introductory section.)

p.9 Eq.3: Even if it is obvious for most readers, N should be explicitly defined.

p.9 Tab.3: That could be helpful to include the vertical resolution and the altitude range covered by each of these instruments.

p.9 l.140: Regarding the MLS vertical resolution, you should explicitly say that it is of the same order as that of SMILES, in order to highlight the fact that the observed differences in the profiles are not due to differences in vertical resolution. (Same comment about the comparison with ACE in Sect. 4.2.)

p.10 l.148-149: There are however changes with latitude observed below 35 km. Please describe them.

p.10 l.155-156: The water vapor effect should be explained (maybe not here, but earlier in the paper, when explaining the retrieval process).

p.10 l.166: "MAD" has not been defined.

p.12 l.192: Please specify the geographic coordinates of the Kiruna station.

p.13 Fig.7: It is good that information about the variability of the differences is given in panels (B), by the representation of $\pm 1\ \sigma$. However, I wonder why this information is not given for difference profiles shown in panels (A), as well as in Fig. 9 (A).

p.18 Eq.10: I guess that $\Delta b0$ is the uncertainty on model parameters. Please define it explicitly.

p.18 l.234: "about 0.9 ppbv at 50 km" This value is inconsistent with what is shown in Fig. 11. Please correct.

p.18-20, Sect. 5.2: Please give more information about the temperature data used in the retrieval process, for the three instruments under consideration. Has the temperature been retrieved from measurements performed by the instrument itself or has, in some cases, external data been used? Discuss the quality of these T data sets, comment on their accuracy. Are there some validation studies that could give an indication as to which ones of the SMILES, MLS and ACE-FTS temperature profiles are closer to the true atmospheric temperature? Such additional information would be helpful for future users to know which of these three HCl data sets is likely the most realistic in the upper stratosphere / lower mesosphere. Knowing more about the temperature data used in the SMILES retrieval procedure would also be useful to better estimate the quality of the SMILES HCl data set in higher altitude regions, where measurements from other instruments are not available.

p.21, Fig.12: The quality of this figure needs to be improved. The legends are barely readable. It is confusing that panel (A) does not have the same vertical scale as the other ones. Also, it would be clearer to use the same colour code or line styles in both panels (C) and (D).

Technical corrections

p.1 l.2: Change "has been" to "is".

p.1 l.12: Change "well agreed" to "agreed well".

p.1 l.18: "concentration" add an "s".

p.2 l.30: "HALOE HCl" remove "HCl".

p.2 l.48-49: Incomplete sentence (no verb).

p.3 l.58: "observation" add an "s".

p.3 l.60: Reword (suggestion "SMILES operational period started on October 12, 2009 and ended on April 21, 2010.")

p.3 l.4: "observation" add an "s". "Kasai et al. (2013)" add "by Kaisai et al... Âż

p.4 l. 68: Change "and" to "or".

p.4 l.92: "y is THE observed spectrum".

p.6 Fig.2, caption: Change "spectrum" to "spectra" (three times). (Same comment about the caption of Fig. 3.)

p.6 l.105: "the altitudeS 50 km-90 km" or "the altitude range 50 km-90 km".

p.6 l.110-112: Reword (suggestion "The HCl vertical distribution shows an increase with altitude with a maximum below the stratopause, approximately constant values between [. . .], and a decrease with altitude from the mesopause to. . .")

p.6 l.114: "panelS (B)"

p.7 l.122: "observationS"

p.8 Fig.4, caption: ". . . within latitude bins of 10°."

p.9 l.132: "previous work ON MLS observations".

p.10 l.147: "increases" remove the "s".

p.10 l.154: "at below 30 km" remove "below".

p.10 l.156: Change "was" to "is".

p.10 l.156: "one of the possible results" Do you mean "one of the possible causes"?

p.10 l.157&170: Change "less" or "lower".

p.10 l.169: "945" There is a mistake. This value is different from the one given in Table 3.

p.10 l.172: Change "tropic region" to "tropical region".

p.10 l.173: "altitude" written twice in a row.

p.11 l.177: "conformed" Do you mean "confirmed"?

p.12 Fig.6: Adding "SMILES" and "MLS" as a title for the left and middle panels would make the figure clearer.

p.13 Fig.7, caption: Change "Eq (4)" to "Eq (3)".

p.18 l.226: "valueS"

p.18 l.234: Change "between the altitude region of 30 and 60 km" to "between 30 and 60 km".

p.19 l.249: "synthesizes" reword. "lower smaller" remove smaller.

p.20 l.269: Change "a had" to "had a".

p.20 l.270: Change "were" to "was".

p.20 l.273-275: Reword (see previous comment about l.110-112).

p.20 l.281: "coincidenceS"

p.21 Fig.12, caption: Change "dash" to "dashed"

p.21 l.284: Change "The negative bias" to "A negative bias".

p.21 l.295: "improvement of THE retrieval algorithm".

———————————————

---

## Author Comment (AC2) · 14 Aug 2020

**Reply to Anonymous Referee #1**

**General comments from Anonymous Referee #1**
The paper presents an important new data set for middle atmospheric studies. The retrieval process is presented clearly, and the validation method is valid. The overall presentation is well structured. However, some important explanation is missing. For example, you say that the water vapor effect is a possible cause for the

observed latitudinal differences, but you do not explain anything about this effect, or you describe the HCl vertical and geographical distribution, without discussing the chemical and dynamical mechanisms controlling it. Last but not least, you conclude that the temperature data used to retrieve HCl is responsible for up to 30 to 40% of the observed differences between SMILES and the other instruments, but you do not give any information about the source and quality of these data sets. Please see my specific comments below for more detailed questions and suggestions. Moreover, the language should be improved. See my suggestions for the technical corrections below.

**Author's response**
Dear Anonymous Referee #1

Thank you very much for your cooperation to improve our manuscript.
We answered all your comments improved our manuscript as follows.

Major improvements are:
· the description of water vapor effect: We removed the description of the water vapor effect because it was not verified enough. Please see the Author's response (1-12).
· the details of HCl distribution: We added a note on the chemical and physical mechanism that produces the HCl vertical and geographical distribution. Please see the Author's response (1-7).
· the details of the a priori temperature profiles: We added a note that the details of the a priori temperature profile used in SMILES, MLS, and ACE-FTS retrieval. Please see the Author's response (1-18).

Also, the English language corrections you pointed out were revised throughout the text. Please find the supplement pdf file of our answer to your comments and the revisions according to your suggestions.
We hope that our manuscript is suitable for the publication in AMT.

Sincerely.

Seidai Nara[1,2], Yasuko Kasai[1,2]

[1]National Institute of Information and Communications Technology
[2]University of Tsukuba

Please also note the supplement to this comment:
https://amt.copernicus.org/preprints/amt-2020-105/amt-2020-105-AC2-supplement.pdf
* * *

---

## Author Response (AR1)

**Point-By-Point Reply to Referee Comment 1 from Anonymous Referee #2**

**Detailed comments**

**Reviewer Comment (1)**
**- Abstract, last sentence: The last sentence of the abstract should be moved further up, somewhere in the first third of the abstract. Acronyms should be introduced also in the abstract.**

**Author's response (1)**
Thank you for pointing it out. We moved the last sentence to Line 4 – 6 and we introduced acronyms in the abstract.

**Author's changes in the manuscript (1)**
-Page 1, Line 4–6 :
Moved "The HCl vertical profile from the upper troposphere to the lower thermosphere is reported for the first time from SMILES observations; the data quality is quantified by comparison with other measurements and via theoretical error analysis" from line 24 – 25 to line 4 – 6.

-Page 1, Line 11
"SD-WACCM model" → "Whole Atmosphere Community Climate Model in specified dynamics configuration (SD-WACCM.)"

-Page 1, Line 12, 13:
"MLS" → "Microwave Limb Sounder (MLS)"

-Page 1, Line 13, 14:
"ACE-FTS" → "Atmospheric Chemistry Experiment - Fourier Transform Spectrometer (ACE-FTS)"

-Page 1, Line 14:
"TELIS" → "TErahertz and submillimeter LImb Sounder (TELIS)"

**Reviewer Comment (2)**
**-Line 86: This is the location where a description of the temperature retrieval should be provided. It will set the stage for a lot of the discussion further below. It should answer questions such as: How is the temperature retrieved? What is the vertical range of the temperature retrieval? How accurate is it, especially in the mesosphere?**

**Author's response (2)**
We are grateful for your valuable comment. The temperature was fixed and not retrieved for the HCl retrieval in the SMILES version 3.0.0 retrieval. The MLS measurements and GEOS-5 model output were combined to synthesize the temperature profile used in the HCl retrieval (Kuribayashi et al., 2017). The accuracy of the temperature profile depends on GEOS5 and MLS, not on SMILES temperature observations. The temperature profile was retrieved using the ozone transition, but was not applied in the HCl retrieval process (SMILES L2r User Guide v3.0.0). We improved our manuscript to make this point clear as follows.

**Author's changes in the manuscript(2)**
-Page 4, Line 93 – 99:
Added " The temperature profile used in the SMILES version 3.0.0 retrieval was synthesized, assuming the hydrostatic equilibrium, using the Goddard Earth Observing System, Version 5 (GEOS-5) reanalysis meteorological datasets and the climatology based on the Aura/MLS measurements (Kuribayashi et al., 2017). The

GEOS-5 datasets were used in the upper troposphere and stratosphere, and the Aura/MLS datasets were used in the mesosphere and lower thermosphere, respectively. The temperature profile was also retrieved using the ozone transition in the SMILES version 3.0.0 retrieval process, but it was not applied to the retrieval of atmospheric species including HCl, except for ozone, to avoid systematic error propagation issues (SMILES-NICT, 2014)."

**Reviewer Comment (3)**
**- Line 110: "an increasing with the altitude increased" should be "an increase with altitude"**

**Author's response (3)**
Thank you for your comment. We improved the manuscript following your comment and comment given by the Anonymous Referee #1 (NUMBER).

**Author's changes in the manuscript(3)**
- Page 7, Line 124, 125 & Page 21, Line 326, 327:
"profile showed increasing with the altitude increased" → "distribution shows an increase with altitude with a maximum"

**Reviewer Comment (4)**
**- Line 111: "decreased with the altitude increasing" should be "decreased with altitude"**

**Author's response (4)**
We revised our manuscript as follows.

**Author's changes in the manuscript (4)**
-Page 7, Line 126 & Page 21, Line 328:
"decreased with the altitude increasing" → "a decrease with altitude"

**Reviewer Comment (5)**
**- Line 116 concerning differences between SMILES and WACCM in the mesosphere: Is there an explanation for these differences? How does the high HCl near the mesopause compare with Cly at lower altitudes? Is this realistic? A depiction of the HCl/Cly ratio based SMILES HCl and Cly estimates from lower altitudes in comparison to the WACCM HCl/Cly ratio may be interesting.**

**Author's response (5)**
We appreciate you pointing it out. The difference of the HCl VMR between SMILES and WACCM is $0.4 - 1.0$ ppbv in the upper mesosphere and lower thermosphere. The difference is larger than the total systematic error in the SMILES HCl VMR estimated by the quantitative error analysis ($0.1 - 0.4$ ppbv), thus, this difference is significant. It might be caused by a combination of the SMILES observation and the WACCM model uncertainties. The underestimation of the model could be because of uncertainties in the HCl photodissociation and the reaction with OH radical, which are dominant destruction mechanisms of HCl for the mesosphere/thermosphere (Brasseur and Solomon, 2005). Also, the $HCl/Cl_y$ ratio is also a quite interesting point as you mentioned. But $HCl/Cl_y$ is out of focus since the subject of this paper is just validation.

**Author's changes in the manuscript (5)**
- Page 8, Line $145 - 150$ : Added "At $70 - 100$ km, the difference of the HCl VMR between SMILES and WACCM in the upper mesosphere and lower thermosphere is $0.4 - 1.0$ ppbv, which is larger than the total systematic error in the SMILES HCl VMR estimated by quantitative error analysis ($0.1 - 0.4$ ppbv).

Thus, the difference is significant. It might be caused by a combination of the SMILES observation and the WACCM model uncertainties. The underestimation of the model could be because of uncertainties in the HCl photodissociation and the reaction with OH radical, which are dominant destruction mechanisms of HCl for the mesosphere/thermosphere (Brasseur and Solomon, 2005). "

**Reviewer Comment (6)**
**- Lines 138-139: How can the accuracy be smaller than the precision? In my understanding the precision is largely due to statistical errors, while the accuracy also includes systematic errors and hence should be larger than the precision. This should be corrected or explained.**

**Author's response (6)**
Thank you for your comment. Precision in this framework is attributed to random sources of error (e.g., radiance errors), whereas accuracy is attributed to systematic errors only. Therefore, accuracy can indeed be smaller than precision. Here we referred the numbers of precision and accuracy from the document of the MLS Version 4.2x Level 2 data quality and description document (Livesey et al., 2018). In this document, precision and accuracy are defined as 0.8 and 0.3 ppbv, respectively.

**Reviewer Comment (7)**
**- Lines 155-156: The discussion of water vapor as a source for differences is not convincing. There is very little water vapor in the stratosphere around 30 km.**

**Author's response (7)**
We appreciate your valuable point of view. We agreed with your suggestion and have removed the statement as follows. As for the discussion of the water vapor effect, we removed it because it was not verified enough. The reason for describing water vapor is the treatment of the continuum in the SMILES retrieval algorithm. In the SMILES retrieval algorithm, we do not retrieve the continuum simultaneously when deriving the species abundance, but rather we give it as a parameter. The continuum can also be seen in observed spectra at about 30 km in Fig.2. The effect of this continuum is also mentioned in the SMILES ozone validation paper (Kasai et al., 2013). On these grounds, we described it as a water vapor effect, but due to a lack of quantitative discussion, we removed it in this paper. We're glad you pointed that out.

**Author's changes in the manuscript (7)**
- Page 10, Line 184 – 186:
Removed "There is a possibility that this difference was caused by water vapor. The SMILES HCl profile was retrieved without considering water vapor effect and the influence of the water vapor was thus one of the possible results of the latitudinal difference."

**Reviewer Comment (8)**
**- Line 166: What is "MAD"? Abbreviations should be introduced.**

**Author's response (8)**
MAD is the acronym for Median Absolute Deviation.

**Author's changes in the manuscript (8)**
- Page 12, Line 198:
"3-MAD" → "3 times the median absolute deviation (3-MAD)"

[Figure]

Figure 1: H$^{35}$Cl spectra observed by SMILES.

**Reviewer Comment (9)**
**- Figure 5: For the right panel an x-axis range at which relative difference does not go off scale should be used.**

**Author's response (9)**
Thank you for pointing it out. We revised that according to your advice.

**Author's changes in the manuscript (9)**
-Page 11, Figure 5, Right panel: We extended the range of the x-axis.

**Reviewer Comment (10)**
**- Line 176: "overlapped" → "overlap"**

**Author's response (10)**
Thank you for your advice. We revised that according to your advice.

**Author's changes in the manuscript (10)**
-Page 12, Line 209:
"overlapped" → "overlap"

**Reviewer Comment (11)**
**- Line 177: "conformed" → "confirmed"**

**Author's response (11)**

Thank you for pointing it out. We revised that according to your advice.

**Author's changes in the manuscript (11)**
-Page 12, Line 210:
"comformed" → "confirmed"

**Reviewer Comment (12)**
**- Line 199: The reference to Webster is ok for the mechanism but is there also a reference to the development of chlorine activation for that particular winter. That would be more relevant.**

**Author's response (12)**
We appreciate you pointing it out. We included Wegner et al., (2016) as a reference. This paper refers to chlorine activation in the Arctic winter of 2009/10 seasons.

**Author's changes in the manuscript (12)**
-Page 14, Line 233:
"Webster et al., 1993" → "Webster et al., (1993); Wegner et al., (2016)"

**Reviewer Comment (13)**
**- Line 201-202: Is there a reference that supports this statement about calibration uncertainties and their impact on the TELIS retrieval?**

**Author's response (13)**
Thank you for your comment. We included Xu et al., (2018) as a reference. This paper refers to TELIS observation.

**Author's changes in the manuscript (13)**
-Page 14, Line 236:
Added "(Xu et al., 2018)"

**Reviewer Comment (14)**
**- Figure 10: Same comment as for figure 5 for the right panel an x-axis range at which relative difference does not go off scale should be used.**

**Author's response (14)**
Thank you for your advice. We revised that according to your advice.

**Author's changes in the manuscript (14)**
-Page 16, Figure 10, Right panel:
We extended the range of the x-axis.

**Reviewer Comment (15)**
**- Table 4: I don't understand the statement in the footnote. The comparison didn't actually look that bad at these altitudes.**

**Author's response (15)**
We appreciate you pointing it out. As you pointed out, it made little sense. Thus, we removed this statement.

**Author's changes in the manuscript (15)**
-Page 17,Table 4: Remove footnote statements.

**Reviewer Comment (16-1)**
**- Line 242: The paper states that "The temperature profile used for the retrieval procedure of SMILES was smaller than those of MLS and ACE-FTS particularly in the upper stratosphere, mesosphere, and lower thermosphere", however, no temperatures for upper mesosphere, lower thermosphere are shown. This should be added, probably in figure 12.**

**Author's response (16-1)**
Thank you for pointing it out. In this section, We focus our discussion on the altitude range, which allows for comparison with other satellite observation values. Thus, we removed the mention of the lower thermosphere, as this was misleading.

**Author's changes in the manuscript (16-1)**
-Page 18, Line 276, 277:
"the upper stratosphere, mesosphere, and lower thermosphere." $\rightarrow$ "the upper stratosphere and mesosphere."

**Reviewer Comment (16-2)**
**- It should be stated how the temperatures for the other satellite measurements were derived.**

**Author's response (16-2)**
We are grateful for your kind advice. The details of the a priori temperature profile for SMILES, MLS and ACE-FTS retrieval have been added. Please also find in the improved manuscript attached.

**Author's changes in the manuscript (16-2)**
-Page 20, Line 289 – 300:
Added "The a priori temperature profile used in the SMILES retrieval procedure is based on the GEOS-5 profile in the stratosphere and MLS retrieved profile above the mesosphere.The altitude limit of the MLS temperature profile is 0.001 hPa, with a vertical resolution of 6 - 14 km and a precision of $1.2 – 3.6$ K per profile. MLS uses GEOS-5 up to 1 hPa as with SMILES. For pressures smaller than 1 hPa, the COSPAR International Reference Atmosphere (CIRA-86) is used as a priori temperature information (with a loose constraint) in the MLS retrieval procedure (Schwartz et al., 2008). The altitude range and vertical resolution of the CIRA-86 profile are ground to 120 km and 2 km, respectively (Fleming et al., 1990). The temperature value retrieved by MLS is 10 K lower than the a priori profile on average in some areas for pressure values smaller than 1 hPa, based on earlier version validation studies (Schwartz et al., 2008). The ACE-FTS retrieval procedure uses the retrieved temperature profile as the a priori between $18 – 125$ km. The vertical resolution of ACE-FTS retrieved temperature is $3 – 4$ km. The temperature values retrieved by ACE-FTS are less than 10 K larger than the MLS derived temperatures. (Schwartz et al., 2008). These types of difference are also seen in the comparison results performed here."

**Reviewer Comment (16-3)**
**- Instead of the word "smaller", I'd maybe suggest "lower" or "colder"**

**Author's response (16-3)**
Thank you for your advice. We revised that according to your advice.

**Author's changes in the manuscript (16-3)**
-Page 18, Line 276 & Page 20, Line 288, 289:

"smaller" → "lower"

**Reviewer Comment (17)**
**- Line 253: "at the altitude (of) 50–70km." – Again, no data is shown for 70 km altitude.**

**Author's response (17)**
We appreciate you pointing it out. We changed the altitude range from "50 – 70 km" to "for altitudes between 50 and 60 km" for the same reason as comment 16-1.

**Author's changes in the manuscript (17)**
-Page 20, Line 289:
"50 – 70 km" → "for altitudes between 50 and 60 km"

**Reviewer Comment (18)**
**- Figure 12: The altitude scale is different in panel a compared to panels b, c, d. These panels should be plotted over the same vertical range avoid confusion. Also, the figure quality is inferior, with the lines and text kind of blurry even when zoomed in, so the figure should be redone.**

**Author's response (18)**
Thank you for pointing it out. We revised the altitude scale that according to your advice. We also increased the resolution of the plot.

**Author's changes in the manuscript (18)**
-Page 21, Figure 12 :
Figure improvements and corrections

**Reviewer Comment (19)**
**- Lines 260-265, discussion of bias at 55 km: What about other altitudes? Figs. 5 and 8 suggest that differences start around 40 km and increase gradually with altitude. However, Fig. 12 suggests that the temperatures around 40 km are very consistent. This behavior should be included in the discussion.**

**Author's response (19)**
We are grateful for your kind advice. As you pointed out, we should have discussed the 40 – 60km altitude range. The behavior of the bias which increased gradually with altitude was also described in this section.

**Author's changes in the manuscript (19)**
-Page 19, Line 301 – 303:
"The difference of the temperature profile caused increase HCl abundance by 0.12 ans 0.20 ppbv at about 55 km for" → "This temperature difference caused an increase in SMILES HCl of 0.12 and 0.20 ppbv at 50 – 60 km for the"

-Page 20, Line 305:
"at 55 km" → "between 40 – 60 km"

-Page 20, Line 308, 309:
"We assumed the 3 % uncertainty in $\gamma_{air}$ and it might cause approximately 0.1 ppbv negative bias at 55 km."
→ "We assumed a 3 % uncertainty in $\gamma_{air}$ which could lead to a −0.1 ppbv bias in the 40 – 60 km region."

-Page 20, Line 309 – 311:
Added "In addition to the error due to the $\gamma_{air}$ coefficients, the effect of temperature differences should be taken into account at altitudes above 50 km. The gradual increase in bias is caused by the difference in altitude at which these two errors become more pronounced. "

-Page 20, Line 312, 313:
"approximately 0.2 ppbv negative bias" $\rightarrow$ "a negative bias of about 0.2 ppbv at 50-60 km."

-Page 20, 21, Line 319 – 323:
"The 0.15 in 0.4 ppbv (approximately 40 %) negative bias between SMILES and MLS was explained. Conclusively, our theoretical error analysis showed that the SMILES HCl a had negative bias of 0.2–0.25 ppbv at 55 km which were consistent with the difference from the MLS and ACE-FTS within the 1 $\sigma$ standard deviation."
$\rightarrow$ "About 40 % (0.15 out of 0.4 ppbv) of the negative bias between SMILES and MLS can beexplained. Therefore, our theoretical error analysis shows that the SMILES HCl has a negative bias of at most 0.25 ppbv be-tween 40 and 60 km; remaining difference between SMILES and MLS or ACE-FTS can be explained by the standard deviation in the comparison result. "

-page 22, Line 348:
"the negative bias of 0.12 and 0.20 ppbv at about 55 km" $\rightarrow$ "a negative bias of 0.12 to 0.20 ppbv between 50 – 60 km"

-Page 22, Line 352 – 354:
"our theoretical error analysis showed that the SMILES HCl had negative bias of 0.20–0.25 ppbv at 55 km which were cons stent with the difference from the MLS and ACE-FTS within the 1 $\sigma$ standard deviation."
$\rightarrow$ "our theoretical error analysis showed that the HCl profiles had a negative bias of 0.20 – 0.25 ppbv at 50 – 60 km, which is consistent with the observed differences versus MLS and ACE-FTS profiles within 1 standard deviation."

**Reviewer Comment (20)**
**- Line 292: "Totally," – I suggest maybe "In Summary,"**

**Author's response (20)**
Thank you for pointing it out. We revised that according to your advice.

**Author's changes in the manuscript (20)**
-Page 22, Line 352:
"Totally" $\rightarrow$ "In summary"

**Reviewer Comment (21)**
**- Line 295: "above stratosphere" – I suggest "above the stratopause"**

**Author's response (21)**
Thank you for your comment. We revised that according to your advice.

**Author's changes in the manuscript (21)**
-Page 22, Line 355:
"above stratosphere" $\rightarrow$ "above the stratopause"

**Point-By-Point Reply to Referee Comment 2 from Anonymous Referee #1**

**1: Specific comments**

**Reviewer Comment (1-1)**
-p.1 l.12-14: Please specify also the relative values corresponding to the given absolute differences.

**Author's response (1-1)**
We appreciate your suggestion. We added the relative values corresponding to the absolute differences.

**Author's changes in the manuscript (1-1)**
-Page 1, Line 16:
Added "(within 10% between 30 and 40 km. There's a larger discrepancy below 30 km.)"

-Page 1, Line 18:
Added "(12 − 15 %)"

**Reviewer Comment (1-2)**
- p.2 l.43-46: Could you please explicitly comment the differences between the results of the comparisons mentioned here (between SMILES v2 and MLS / ACE-FTS, by Sugita et al., 2013) and the results of your validation study? It would be interesting to add such a comment in the conclusion section, where you discuss your results.

**Author's response (1-2)**
Thank you for pointing it out. This earlier paper by Sugita et al., 2013 used the v215 SMILES data and limited to an altitude of around 30 km in the polar region. In contrast, this study used the v300 data and performed global comparison in the stratosphere and mesosphere. We added a note regarding the features of this paper.

**Author's changes in the manuscript (1-2)**
-Page 2, 3, Line 51, 53:
Added "In contrast to the previous study, this study used v300 SMILES data and provided a global comparison, including the stratosphere and mesosphere."

**Reviewer Comment (1-3)**
- p.4. l.76: Do not forget to specify in the text that you are talking about the daily number of observations.

**Author's response (1-3)**
Thank you for your comment. We mentioned that we are talking about the number of observations per day.

**Author's changes in the manuscript (1-3)**
-Page 3, Line 83:
"the number of observations" → "the number of observations per day"

**Reviewer Comment (1-4)**

**- p.4 l.92: Could you give some information about the a priori data used in the retrieval process: what is the source of this data set? Does it depend on latitude and time? Etc.**

**Author's response (1-4)**
Thank you for pointing it out. The U.S. standard atmosphere is used as the a priori profile in SMILES version 3.0.0 retrieval. And the a priori profile is used separately in the polar, equatorial, mid-Summer, and mid-winter regions. We described the treatment of the a priori profile in the manuscript.

**Author's changes in the manuscript (1-4)**
-Page 6, Line 106 – 108:
Added "We used the U.S. standard atmosphere profiles as the a priori state (xa). They are used separately for polar, equatorial, summer mid-latitude, or winter mid-latitude regions (US-Standard 1976)."
(US-Standard, P.: U.S Standard Atmosphere 1976, Tech. rep., U.S Government Printing Office Washington DC, 1976)

**Reviewer Comment (1-5)**
**- p.6 l.102: "We assumed that the natural isotopic abundance of H35Cl/HCl was 0.7576. . .":
Please explain where this value comes from. A citation should be added here.**

**Author's response (1-5)**
We appreciate you pointing it out. We added Berglund and Wieser (2011) to the citation.

**Author's changes in the manuscript (1-5)**
-Page 7, Line 116:
"was 0.7576" $\rightarrow$ "was 0.7576 by Berglund and Wieser., 2011"
(Berglund, M. and Wieser, M. E.: Isotopic compositions of the elements 2009 (IUPAC Technical Report), Pure and Applied Chemistry, 83, 397360– 410, https://doi.org/https://doi.org/10.1351/PAC-REP-10-06-02, 2011)

**Reviewer Comment (1-6)**
**- p.6-7 Sect.3: Please discuss the differences and/or similarities between the three subperiods under consideration or change Fig. 4 to show only the results averaged over the whole SMILES operational period. I do not understand the point in dividing the comparisons between SMILES and SD-WACCM into three different time periods if this is not discussed.**

**Author's response (1-6)**
We are grateful for your kind comments. We changed Fig. 4 to show only the results averaged throughout the SMILES operational period.

**Author's changes in the manuscript (1-6)**
-Page 9, Figure 4: The figure was improved.

-Page 9, Figure 4, caption:
"rows" $\rightarrow$ "panel"

Removed "in three time range"

- Page 7, Line 119, 120:
"Left panel: Time range between October 16, 2009 and December 15, 2009. Middle panel: Between December 16, 2009 and February 15, 2010. Right panel: Between February 16, 2010 and April 17, 2010."

$\rightarrow$ "The time period is between October 16, 2009 and April 17, 2010."

**Reviewer Comment (1-7)**
**- p.6 l.110-112: An explanation about the HCl vertical and latitudinal distribution is missing. Please describe the chemical and physical mechanisms controlling it, or at least comment on the current state of knowledge about that. (This could be added either here or in the introductory section.)**

**Author's response (1-7)**
We appreciate you pointing it out. We explained the chemical and physical mechanism that produced the distribution of HCl.

**Author's changes in the manuscript (1-7)**
-Page 7,8 , Line 127 – 131:
Added "In the lower and middle stratosphere, HCl is generated by the reaction of Cl with $CH_4$ and $HO_2$ and transported by circulation (e.g. Brewer-Dobson circulation). The HCl abundance is balanced by production $(Cl + HO2 \rightarrow HCl + O_2)$ and loss $(HCl + OH \rightarrow Cl + H_2O, HCl + h\nu \rightarrow H + Cl)$ in the upper stratosphere and mesosphere. Near the mesopause, the photodissociation becomes the dominant reaction, and the HCl abundance decreases with height (Brasseur and Solomon, 2005)."

**Reviewer Comment (1-8)**
**- p.9 Eq.3: Even if it is obvious for most readers, N should be explicitly defined.**

**Author's response (1-8)**
Thank you for pointing it out. We defined "N" as the number of coincidence.

**Author's changes in the manuscript (1-8)**
-Page 9, Line 157:
Added "N(z) is the number of coincidences at an altitude z"

**Reviewer Comment (1-9)**
**- p.9 Tab.3: That could be helpful to include the vertical resolution and the altitude range covered by each of these instruments.**

**Author's response (1-9)**
Thank you for your comment. We included the vertical resolution and the altitude range of each instruments in Table 3.

**Author's changes in the manuscript (1-9)**
-Page 10, Table 3:
Added the row of "vertical resolution" and "altitude range".

**Reviewer Comment (1-10)**
**- p.9 l.140: Regarding the MLS vertical resolution, you should explicitly say that it is of the same order as that of SMILES, in order to highlight the fact that the observed differences in the profiles are not due to differences in vertical resolution. (Same comment about the comparison with ACE in Sect. 4.2.)**

**Author's response (1-10)**
Thank you for pointing it out. We mentioned the vertical resolution to highlight the fact that the observed differences in the profiles are not due to differences in vertical resolution. Also, we added a reference to ACE-FTS version 4.0 retrieval.

**Author's changes in the manuscript (1-10)**
-Page 10, Line 169, 170:
Added "The vertical resolution of MLS is of the same order as that of SMILES."

-Page 11, Line 195:
Added "(Boone et al., 2020)"
(Boone, C., Bernath, P., Cok, D., Jones, S., and Steffen, J.: Version 4 retrievals for the atmospheric chemistry experiment Fouriertransform spectrometer (ACE-FTS) and imagers, Journal of Quantitative Spectroscopy and Radiative Transfer, 247, 106 939,https://doi.org/https://doi.org/10.1016/j.jqsrt.2020.106939, http://www.sciencedirect.com/science/article/pii/S0022407319305916, 2020.)

-Page 12, Line 197:
Added "The vertical resolution of ACE-FTS is of the same order as that of SMILES."

**Reviewer Comment (1-11)**
**- p.10 l.148-149: There are however changes with latitude observed below 35 km. Please describe them.**

**Author's response (1-11)**
Thank you for your comment. We added a description on this.

**Author's changes in the manuscript (1-11)**
-Page 10, Line 178, 179:
Added "While below 35 km of altitude, several areas of large differences can be identified, especially in the equatorial region."

**Reviewer Comment (1-12)**
**- p.10 l.155-156: The water vapor effect should be explained (maybe not here, but earlier in the paper, when explaining the retrieval process).**

**Author's response (1-12)**
We appreciate your valuable point of view. We agreed with your suggestion and removed the statement as follows. As for the discussion of the water vapor effect, we removed it because it was not verified enough. The reason for describing water vapor is the treatment of the continuum in the SMILES retrieval algorithm. In the SMILES retrieval algorithm, we do not retrieve the continuum simultaneously when deriving the species abundance, but rather we give it as a parameter. The continuum can also be seen in observed spectra at about 30 km in Fig.2. The effect of this continuum is also mentioned in the SMILES ozone validation paper (Kasai et al., 2013). On these grounds, we described it as a water vapor effect, but due to a lack of quantitative discussion, we removed it in this paper. We're glad you pointed that out.

**Author's changes in the manuscript (1-12)**
-Page 10, Line 184 – 186:
Removed "There is a possibility that this difference was caused by water vapor. The SMILES HCl profile

[Figure]

Figure 2: H$^{35}$Cl spectra observed by SMILES.

was retrieved without considering water vapor effect and the influence of the water vapor was thus one of the possible results of the latitudinal difference."

**Reviewer Comment (1-13)**
**- p.10 l.166: "MAD" has not been defined.**

**Author's response (1-13)**
We are grateful for your kind advice. MAD is the acronym for Median Absolute Deviation.

**Author's changes in the manuscript (1-13)**
- Page 12, Line 198:
"3-MAD" $\rightarrow$ "3 times the median absolute deviation (3-MAD)

**Reviewer Comment (1-14)**
**- p.12 l.192: Please specify the geographic coordinates of the Kiruna station.**

**Author's response (1-14)**
Thank you for your comment. We added the information about the geographic coordinates of the Kiruna station.

**Author's changes in the manuscript (1-14)**
-Page 16, Line 225, 226:
"in terms of geolocation and time on January 24, 2010 over Kiruna." $\rightarrow$ "in terms of geolocation over Kiruna (67.8 N° 20.4 E°) and time on January 24, 2010."

**Reviewer Comment (1-15)**
- p.13 Fig.7: It is good that information about the variability of the differences is given in panels (B), by the representation of $\pm 1 \sigma$. However, I wonder why this information is not given for difference profiles shown in panels (A), as well as in Fig. 9 (A).

**Author's response (1-15)**
We are grateful for your kind advice. We added the representation of $\pm 1 \sigma$ in panels (A) according to your comment.

**Author's changes in the manuscript (1-15)**
-Page 13, Figure 7, panels (A) & -Page 15, Figure 9, panels (A) :
Figure corrections and changes.

**Reviewer Comment (1-16)**
- p.18 Eq.10: I guess that $\Delta b_0$ is the uncertainty on model parameters. Please define it explicitly.

**Author's response (1-16)**
Thank you for pointing it out. As you mentioned, $\Delta b_0$ is the uncertainty on parameters. We defined it in the manuscript.

**Author's changes in the manuscript (1-16)**
-Page 18, Line 258:
"where $I$ is the inversion function and $b_0$ is the vector of model parameters." $\rightarrow$ "where $I$ is the inversion function, $b_0$ is the vector of model parameters and $\Delta b_0$ is the uncertainty on model parameters"

**Reviewer Comment (1-17)**
- p.18 l.234: "about 0.9 ppbv at 50 km" This value is inconsistent with what is shown in Fig. 11. Please correct.

**Author's response (1-17)**
We corrected the value to 0.09 ppbv following your comment.

**Author's changes in the manuscript (1-17)**
-Page 18, Line 268:
"about 0.9 ppbv" $\rightarrow$ "about 0.09 ppbv"

**Reviewer Comment (1-18)**
- p.18-20, Sect. 5.2: Please give more information about the temperature data used in the retrieval process, for the three instruments under consideration. Has the temperature been retrieved from measurements performed by the instrument itself or has, in some cases, external data been used? Discuss the quality of these T data sets, comment on their accuracy. Are there some validation studies that could give an indication as to which ones of the SMILES, MLS and ACE-FTS temperature profiles are closer to the true atmospheric temperature? Such additional information would be helpful for future users to know which of these three HCl data sets is likely the most realistic in the upper stratosphere / lower mesosphere. Knowing more about the temperature data used in the SMILES retrieval procedure would also be useful to better estimate the quality of the SMILES HCl data set in higher altitude regions, where

**measurements from other instruments are not available.**

**Author's response (1-18)**
We are grateful for your kind advice. The details of the a priori temperature profile for SMILES, MLS and ACE-FTS retrieval were added.

**Author's changes in the manuscript (1-18)**
-Page 4, Line 93 – 99 :
Added "The temperature profile used in the SMILES version 3.0.0 retrieval was synthesized, assuming the hydrostatic equilibrium, using the Goddard Earth Observing System, Version 5 (GEOS-5) reanalysis meteorolog-ical datasets and the climatology based on the Aura/MLS measurements (Kuribayashi et al., 2017). The GEOS-5 datasets were used in the upper troposphere and stratosphere, and the Aura/MLS datasets were used in the mesosphere and lower thermosphere, respectively. The temperature profile was also retrieved using the ozone transition in the SMILES version 3.0.0 retrieval process, but it was not applied to the retrieval of atmospheric species including HCl, except for ozone, to avoid systematic error propagation issues (SMILES-NICT, 2014)."

-Page 20, Line 289 – 300:
Added "The a priori temperature profile used in the SMILES retrieval procedure is based on the GEOS-5 profile in the stratosphere and MLS retrieved profile above the mesosphere.The altitude limit of the MLS temperature profile is 0.001 hPa, with a vertical resolution of 6 - 14 km and a precision of $1.2 – 3.6$ K per profile. MLS uses GEOS-5 up to 1 hPa as with SMILES. For pressures smaller than 1 hPa, the COSPAR International Reference Atmosphere (CIRA-86) is used as a priori temperature information (with a loose constraint) in the MLS retrieval procedure (Schwartz et al., 2008). The altitude range and vertical resolution of the CIRA-86 profile are ground to 120 km and 2 km, respectively (Fleming et al., 1990). The temperature value retrieved by MLS is 10 K lower than the a priori profile on average in some areas for pressure values smaller than 1 hPa, based on earlier version validation studies (Schwartz et al., 2008). The ACE-FTS retrieval procedure uses the retrieved temperature profile as the a priori between $18 – 125$ km. The vertical resolution of ACE-FTS retrieved temperature is $3 – 4$ km. The temperature values retrieved by ACE-FTS are less than 10 K larger than the MLS derived temperatures. (Schwartz et al., 2008). These types of difference are also seen in the comparison results performed here.".

**Reviewer Comment (1-19)**
**- p.21, Fig.12: The quality of this figure needs to be improved. The legends are barely readable. It is confusing that panel (A) does not have the same vertical scale as the other ones. Also, it would be clearer to use the same colour code or line styles in both panels (C) and (D).**

**Author's response (1-19)**
Thank you for your comment. We revised the altitude scale and improved the figure resolution that according to your comment.

**Author's changes in the manuscript (1-19)**
-Page 21, Figure 12 :
Figure improvements and corrections

**2: Technical corrections.**

**Reviewer Comment (2-1)**

**- p.1 l.2: Change "has been" to "is".**

**Author's response (2-1)**
Thank you for pointing it out. We revised that according to your advice.

**Author's changes in the manuscript (2-1)**
-Page 1, Line 2:
"has been" → "is"

**Reviewer Comment (2-2)**
**- p.1 l.12: Change "well agreed" to "agreed well".**

**Author's response (2-2)**
Thank you for your comment. We revised that according to your comment.

**Author's changes in the manuscript (2-2)**
-Page 1, Line 16:
"well agreed" → "agreed well".

**Reviewer Comment (2-3)**
**- p.1 l.18: "concentration" add an "s".**

**Author's response (2-3)**
Thank you for your comment. We revised that according to your advice.

**Author's changes in the manuscript (2-3)**
-Page 2, Line 23:
"concentration" → "concentrations".

**Reviewer Comment (2-4)**
**- p.2 l.30: "HALOE HCl" remove "HCl".**

**Author's response (2-4)**
We appreciate you pointing it out. We revised that according to your comment.

**Author's changes in the manuscript (2-4)**
-Page 2, Line 35:
"HALOE HCl" → "HALOE"

**Reviewer Comment (2-5)**
**- p.2 l.48-49: Incomplete sentence (no verb).**

**Author's response (2-5)**
Thank you for pointing it out. We revised that according to your advice.

**Author's changes in the manuscript (2-5)**
-Page 3, Line 54:
"SMILES observation of" → "we examined"

**Reviewer Comment (2-6)**
**- p.3 l.58: "observation" add an "s".**

**Author's response (2-6)**
Thank you for your comment. We revised that according to your advice.

**Author's changes in the manuscript (2-6)**
-Page 3, Line 64:
"observation" → "observations"

**Reviewer Comment (2-7)**
**- p.3 l.60: Reword (suggestion "SMILES operational period started on October 12, 2009 and ended on April 21, 2010.")**

**Author's response (2-7)**
We are grateful for your kind advice. We revised that according to your advice.

**Author's changes in the manuscript (2-7)**
-Page 3, Line 66, 67:
"The period of the SMILES observation was from October 12, 2009 to April 21, 2010." → "SMILES operational period started on October 12, 2009 and ended on April 21, 2010."

**Reviewer Comment (2-8)**
**- p.3 l.4: "observation" add an "s". "Kasai et al. (2013)" add "by Kaisai et al. . . "**

**Author's response (2-8)**
We appreciate you pointing it out. We revised that according to your advice.

**Author's changes in the manuscript (2-8)**
-Page 3, Line 70:
"observation" → "observations"

-Page 3, Line 71:
"Kasai et al. (2013)" → "by Kasai et al. (2013)"

**Reviewer Comment (2-9)**
**- p.4 l. 68: Change "and" to "or".**

**Author's response (2-9)**
Thank you for pointing it out. We revised that according to your advice.

**Author's changes in the manuscript (2-9)**
-Page 3, Line 75:
"and" → "or"

**Reviewer Comment (2-10)**

**- p.4 l.92: "y is THE observed spectrum".**

**Author's response (2-10)**
Thank you for pointing it out. We revised that according to your advice.

**Author's changes in the manuscript (2-10)**
-Page 6, Line 105:
"y is observed spectra" → "y is the observed spectra"

**Reviewer Comment (2-11)**
**- p.6 Fig.2, caption: Change "spectrum" to "spectra" (three times). (Same comment about the caption of Fig. 3.)**

**Author's response (2-11)**
Thank you for your comment. We revised that according to your advice.

**Author's changes in the manuscript (2-11)**
-Page 6, Figure 2, caption:
"spectrum" → "spectra"

**Reviewer Comment (2-12)**
**- p.6 l.105: "the altitudeS 50 km-90 km" or "the altitude range 50 km-90 km".**

**Author's response (2-12)**
We appreciate you pointing it out. We revised that according to your advice.

**Author's changes in the manuscript (2-12)**
-Page 7, Line 119:
"altitude of $50\,\mathrm{km} - 90\,\mathrm{km}$" → "altitudes $50\,\mathrm{km} - 90\,\mathrm{km}$"

**Reviewer Comment (2-13)**
**- p.6 l.110-112: Reword (suggestion "The HCl vertical distribution shows an increase with altitude with a maximum below the stratopause, approximately constant values between [. . .], and a decrease with altitude from the mesopause to. . .")**

**Author's response (2-13)**
We are grateful for your kind comment. We revised that according to your advice.

**Author's changes in the manuscript (2-13)**
-Page 7, 8, Line 124, 125:
"The HCl vertical profile showed an increasing with the altitude increased below the stratopause ($\sim 45$ km), approximately constant between" → "The HCl vertical distribution shows an increase with altitude with a maximum below the stratopause ($\sim 45$ km), approximately constant values between"

-Page 7, Line 126, 127:
"decreased with the altitude increasing" → "a decrease with altitude"

**Reviewer Comment (2-14)**

**- p.6 l.114: "panelS (B)"**

**Author's response (2-14)**
Thank you for pointing it out. We changed the figure 4 (B) to one.

**Reviewer Comment (2-15)**
**- p.7 l.122: "observationS"**

**Author's response (2-15)**
Thank you for pointing it out. We revised that according to your advice.

**Author's changes in the manuscript (2-15)**
-Page 8, Line 144:
"observation" → "observations"

**Reviewer Comment (2-16)**
**- p.8 Fig.4, caption: ". . . within latitude bins of 10°."**

**Author's response (2-16)**
Thank you for your advice. We revised that according to your advice.

**Author's changes in the manuscript (2-16)**
-Page 9, Figure 4, caption:
"a latitude bin of" → "latitude bins of"

**Reviewer Comment (2-17)**
**- p.9 l.132: "previous work ON MLS observations"**

**Author's response (2-17)**
Thank you for pointing it out. We revised that according to your advice.

**Author's changes in the manuscript (2-17)**
-Page 10, Line 161:
"previous work of the MLS observations" → "previous work on MLS observations"

**Reviewer Comment (2-18)**
**- p.10 l.147: "increases" remove the "s".**

**Author's response (2-18)**
Thank you for your comment. We revised that according to your advice.

**Author's changes in the manuscript (2-18)**
-Page 10, Line 176:
"increases" → "increase"

**Reviewer Comment (2-19)**
**- p.10 l.154: "at below 30 km" remove "below".**

**Author's response (2-19)**
We appreciate you pointing it out. We revised that according to your advice.

**Author's changes in the manuscript (2-19)**
-Page 11, Line 184:
"at below 30 km" → "at 30 km"

**Reviewer Comment (2-20)**
**- p.10 l.156: Change "was" to "is".**

**Author's response (2-20)**
Thank you for your advice. We removed the sentence containing this. See Author's response (1-12) for more information.

**Reviewer Comment (2-21)**
**- p.10 l.156: "one of the possible results" Do you mean "one of the possible causes"?**

**Author's response (2-21)**
We appreciate you pointing it out. We removed the sentence containing this. See Author's response (1-12) for more information.

**Reviewer Comment (2-22)**
**- p.10 l.157&170: Change "less" or "lower".**

**Author's response (2-22)**
Thank you for your comment. We revised that according to your advice.

**Author's changes in the manuscript (2-22)**
-Page 11, Line 188 & Page 12, Line 202
"less" → "lower"

**Reviewer Comment (2-23)**
**- p.10 l.169: "945" There is a mistake. This value is different from the one given in Table 3.**

**Author's response (2-23)**
We appreciate you pointing it out. We have revised the values to be correct.

**Author's changes in the manuscript (2-23)**
-Page 12, Line 201:
"945" → "935"

**Reviewer Comment (2-24)**
**- p.10 l.172: Change "tropic region" to "tropical region".**

**Author's response (2-24)**
Thank you for your advice. We revised that according to your advice.

**Author's changes in the manuscript (2-24)**
-Page 12, Line 205:
"tropic region" → "tropical region"

**Reviewer Comment (2-25)**
**- p.10 l.173: "altitude" written twice in a row.**

**Author's response (2-25)**
Thank you for pointing it out. Removed one of the two "altitude".

**Author's changes in the manuscript (2-25)**
-Page 12, Line 205:
"altitude altitude" → "altitude"

**Reviewer Comment (2-26)**
**- p.11 l.177: "conformed" Do you mean "confirmed"?**

**Author's response (2-26)**
Thank you for pointing it out. The correct word is "Confirmed". We fixed it.

**Author's changes in the manuscript (2-26)**
-Page 12, Line 210:
"conformed" → "confirmed"

**Reviewer Comment (2-27)**
**- p.12 Fig.6: Adding "SMILES" and "MLS" as a title for the left and middle panels would make the figure clearer.**

**Author's response (2-27)**
We are grateful for your kind comment. We revised that according to your advice.

**Author's changes in the manuscript (2-27)**
-Page 12, Figure 6: Added a title for panels

**Reviewer Comment (2-28)**
**- p.13 Fig.7, caption: Change "Eq (4)" to "Eq (3)".**

**Author's response (2-28)**
Thank you for pointing it out. We revised that according to your advice.

**Author's changes in the manuscript (2-28)**
-Page 13, Figure 7, caption:
"Eq (4)" → "Eq (3)"

**Reviewer Comment (2-29)**

**- p.18 l.226: "valueS"**

**Author's response (2-29)**
Thank you for your comment. We revised that according to your advice.

**Author's changes in the manuscript (2-29)**
-Page 18, Line 261:
"value" → "values"

**Reviewer Comment (2-30)**
**- p.18 l.234: Change "between the altitude region of 30 and 60 km" to "between 30 and 60 km".**

**Author's response (2-30)**
Thank you for your comment. We revised that according to your advice.

**Author's changes in the manuscript (2-30)**
-Page 18, Line 268:
"between the altitude region of 30 and 60 km" → "between 30 and 60 km"

**Reviewer Comment (2-31)**
**- p.19 l.249: "synthesizes" reword. "lower smaller" remove smaller.**

**Author's response (2-31)**
Thank you for pointing it out. We revised the sentence.

**Author's changes in the manuscript (2-31)**
-Page 19, Line 282 – 285:
"Negative value of the jacobian means that higher temperature synthesizes lower smaller brightness temperature spectrum, thus, increases the HCl abundance in the retrievals calculation to compensate for the underestimation of the synthesized spectrum due to the temperature higher than the true value."
→ " A negative jacobian value means that highertemperatures induce a lower brightness temperature spectrum, thus increasing the HCl abundance in the retrieval to compensate for this underestimation. "

**Reviewer Comment (2-32)**
**- p.20 l.269: Change "a had" to "had a".**

**Author's response (2-32)**
Thank you for your comment. We revised that according to your advice.

**Author's changes in the manuscript (2-32)**
-Page 20, Line 320, 321:
"showed that the SMILES HCl a had" → "shows that the SMILES HCl has a"

**Reviewer Comment (2-33)**
**- p.20 l.270: Change "were" to "was".**

**Author's response (2-33)**
Thank you for pointing it out. We revised the sentence included this point.

**Author's changes in the manuscript (2-33)**
-Page 20, 21, Line 320 – 323:
"our theoretical error analysis showed that the SMILES HCl a had negative bias of 0.2–0.25 ppbv at 55 km which were consistent with the difference from the MLS and ACE-FTS within the $1 \sigma$ standard deviation."
$\rightarrow$ "our theoretical error analysis shows that the SMILES HCl has a negative bias of at most $0.25 \, \text{ppbv}$ between 40 and $60 \, \text{km}$; remaining difference between SMILES and MLS or ACE-FTS can be explained by the standard deviation in the comparison result."

**Reviewer Comment (2-34)**
**- p.20 l.273-275: Reword (see previous comment about l.110-112).**

**Author's response (2-34)**
We are grateful for your kind comment. We revised that according to your advice.

**Author's changes in the manuscript (2-34)**
-Page 21, 22, Line 329 – 333:
Added "In the lower and middle stratosphere, HCl is generated by the reaction of Cl with $CH_4$ and $HO_2$ and transported by circulation (e.g. Brewer-Dobson circulation). The HCl abundance is balanced by production $(Cl + HO2 \rightarrow HCl + O_2)$ and loss$(HCl + OH \rightarrow HCl + H_2O, HCl + h\nu \rightarrow H + Cl)$ in the upper stratosphere and the mesosphere. Above the mesopause, the photodissociation becomes the dominant reaction and the HCl abundance decreases."

**Reviewer Comment (2-35)**
**- p.20 l.281: "coincidenceS"**

**Author's response (2-35)**
Thank you for your comment. We revised that according to your advice.

**Author's changes in the manuscript (2-35)**
-Page 22, Line 338:
"coincidence" $\rightarrow$ "coincidences"

**Reviewer Comment (2-36)**
**- p.21 Fig.12, caption: Change "dash" to "dashed"**

**Author's response (2-36)**
Thank you for your comment. We revised that according to your advice.

**Author's changes in the manuscript (2-36)**
-Page 21, Figure 12, caption:
"dash" $\rightarrow$ "dashed"

**Reviewer Comment (2-37)**
**- p.21 l.284: Change "The negative bias" to "A negative bias".**

**Author's response (2-37)**
Thank you for your comment We revised that according to your advice.

**Author's changes in the manuscript (2-37)**
-Page 22, Line 341:
"The negative bias" → "A negative bias"

**Reviewer Comment (2-38)**
**- p.21 l.295: "improvement of THE retrieval algorithm".**

**Author's response (2-38)**
We appreciate you pointing it out. We revised that according to your advice.

**Author's changes in the manuscript (2-38)**
-Page 22, Line 355, 356:
"further improvement of retrieval algorithm" → "potential improvement in the SMILES retrieval algorithms"

**Other minor changes in the manuscript.**

**Author's changes in the manuscript (1)**
-Page 8, Line 141, 142:
Moved "Differences of HCl abundance between SMILES and SD-WACCM are shown in the panel (C)." from page 8, Line 133, 134 to page 8, Line 141, 142

**Author's changes in the manuscript (2)**
-Page 8, Line 144, 145:
Moved "The SMILES HCl abundance agrees well with SD-WACCM simulation (within 0.1 ppbv) in the stratosphere and middle mesosphere." from page 8, Line 134, 135 to page 8, Line 144, 145

**Author's changes in the manuscript (3)**
-Page 10, Line 180 & 182:
"Panels" → "Panel"

**Author's changes in the manuscript (4)**
-Page 18, Line 273:
"above $40\,\mathrm{km}$" → "between $40-60\,\mathrm{km}$"

**Author's changes in the manuscript (5)**
-Page 19, Line 281:
"with perturbation" → "with a perturbation"

**Author's changes in the manuscript (6)**
-Page 19, Line 281:
Removed "the"

**Author's changes in the manuscript (7)**
-Page 19, Line 286, 287:
"The vertical profile of temperature used for the retrieval procedure of" → "The temperature profiles used in the retrievals by"

**Author's changes in the manuscript (8)**
-Page 20, Line 288:
Removed "the"

**Author's changes in the manuscript (9)**
-Page 20, Line 303:
"MLS and ACE-FTS" → "MLS and ACE-FTS comparisons"

**Author's changes in the manuscript (10)**
-Page 20, Line 307, 308 :
"temperature profile used in retrieval calculation" → "the temperature profiles used in the retrievals"

**Author's changes in the manuscript (11)**
-Page 20, Line 314, 315:
"SMILES and ACE-FTS was explained by the uncertainty in $\gamma_{\mathrm{air}}$ and temperature profile used in the retrievals." → "SMILES and ACE-FTS can be explained by the uncertainty in $\gamma_{\mathrm{air}}$ and the temperature profiles used in the SMILES retrievals."

**Author's changes in the manuscript (12)**

-Page 20, Line 317, 318:
"The 1 % difference of $\gamma_{\mathrm{air}}$ might cause the HCl abundance increase about 0.03 ppbv at 55 km" → "A 1 % difference in $\gamma_{\mathrm{air}}$ might cause the HCl abundance to increase by about 0.03 ppbv"

**Author's changes in the manuscript (13)**
-Page 22, Line 334:
"at altitude 30 to 70 km" → "for altitudes between 30 and 70 km."

**Author's changes in the manuscript (14)**
-Page 22, Line 335, 336:
"the comparison study with other instrument measurements and the theoretical error analysis"
→ "comparisons versus other measurements, and supported by a theoretical error analysis"

**Author's changes in the manuscript (15)**
-Page 22, Line 337:
"with the" → ", versus"

**Author's changes in the manuscript (16)**
-Page 22, Line 340, 341:
Added "the"
"with in the difference of" → "with differences within"

**Author's changes in the manuscript (17)**
-Page 22, Line 342, 343:
"compared to the MLS and ACE-FTS" → "in comparisons versus MLS and ACE-FTS HCl profiles."

**Author's changes in the manuscript (18)**
-Page 22, Line 344, 345:
"We estimated the total error based on the perturbation method and error due to the uncertainty in the atmospheric temperature profile used in the retrieval calculation."
→ "We estimated the total error for SMILES HCl based on the perturbation method and considering the uncertainties in atmospheric temperature profiles used in the retrievals."

**Author's changes in the manuscript (19)**
-Page 22, Line 350, 351:
"the temperature profile were capable of totally contributing up to 40 – 50 % of the negative bias in 50 – 60 km altitudes." →
"the temperature profile are capable of contributing a total of 40 – 50 % of the SMILES HCl negative biases at 50 – 60 km."

**Author's changes in the manuscript (20)**
-Page 22, Line 357, 358:
Removed "quantitative estimations of the"

**Author's changes in the manuscript (21)**
-Page 22, Line 358, 359:
"Further observations and model studies regarding HCl abundance including upper atmosphere are needed to understand the source and sinks"
→ "Further observations and model studies are needed to better understand the sources and sinks, "

**Author's changes in the manuscript (22)**
-Page 14, Fig. 8 & Page 15, Fig. 9 & Page 21, Fig. 12:

We changed the color of the ACE-FTS plot from red to green.

**Author's changes in the manuscript (23)**
-Page 14, Line 212 & Page 14, Fig. 8, caption & Page 15, Fig. 9, caption & Page21, Fig. 12, caption:
"red" → "green"

[revised manuscript text omitted]

---

## Author Response (AR2)

**Dear Editor Dr. Miriam Sinnhubar**

We deeply appreciate for your support to the manuscript. We have corrected the points from you and the referee #2 as follows.

We hope the current manuscript is suitable for publication in AMT.

Sincerely yours.

Seidai Nara[1,2], and Yasuko Kasai[1,2]
1: National Institute of Information and Communication Technology, 2: University of Tsukuba

**Comments from the editor:**

Please consider the comments of the reviewer. Additionally, please check whether there are brackets needed after the Sum sign in Equation 3: shouldn't it be (xc(s)-xs(s))? Or is there only one Smiles profile compared to many profiles from the other instrument?

**Author's response**
Thank you for pointing it out. In addition to your comment, we corrected the description of the equations (3) and (4) in the main manuscript. We revised the manuscript as follows.

**Author's changes in the manuscript**
-Page 8, Line 145 – 151:
The mean absolute difference between SMILES and the other instrument is defined as

$$\Delta_{\mathrm{abs}}(z) = \frac{1}{N(z)} \sum_{i=1}^{N(z)} (x_c(z) - x_s(z)),$$

where $x_s(z)$ and $x_c(z)$ are the HCl volume mixing ratio (VMR) at an altitude $z$ for SMILES and the other instrument, respectively. $N(z)$ is the number of coincidence at an altitude $z$. The mean relative difference (in percentage) is given by

$$\Delta_{\mathrm{rel}}(z) = \frac{1}{N(z)} \sum_{i=1}^{N(z)} \frac{x_c(z) - x_s(z)}{(x_c(z) + x_s(z))/2} \times 100.$$

$\rightarrow$ The mean absolute difference, $\Delta_{\mathrm{abs}}$, for each altitude between SMILES and the other instrument is defined as

$$\Delta_{\mathrm{abs}}(z) = \frac{1}{N(z)} \sum_{i=1}^{N(z)} (x_{c,i}(z) - x_{s,i}(z))$$

where $x_{s,i}(z)$ and $x_{c,i}(z)$ are the HCl volume mixing ratio (VMR) of $i$th coincidence at an altitude $z$ for SMILES and the other instrument, respectively. $N(z)$ is the number of coincidence at an altitude $z$. The mean relative difference in percentage, $\Delta_{\mathrm{rel}}$, for each altitude is given by

$$\Delta_{\mathrm{rel}}(z) = \frac{1}{N(z)} \sum_{i=1}^{N(z)} \frac{x_{c,i}(z) - x_{s,i}(z)}{(x_{c,i}(z) + x_{s,i}(z))/2} \times 100.$$

**Point-By-Point Reply to Referee Comment from Anonymous Referee #2**

**General comments from Anonymous Referee #2**
The manuscript has been thoroughly revised and all referee comments have been addressed appropriately. I only have a couple of minor technical comments.

**Author's response**
Thank you very much for your cooperation to improve our manuscript. We revised the manuscript according to your suggestions as follows.

**Suggestions for revision.**

**Reviewer Comment (1.)**
In the paragraph concerning the temperature measurements from other satellites that has been inserted in Section 5.2, in line 273 "above the mesosphere" should probably be changed to "in the mesosphere and above".

**Author's response (1.)**
Thank you for pointing it out. We revised the manuscript as follows.

**Author's changes in the manuscript (1.)**
-Page 20, Line: 274
"above the mesosphere" → "in the mesosphere and above"

**Reviewer Comment (2.)**
In the same section, I have trouble understanding the statement "The ACE-FTS retrieval procedure uses the retrieved temperature profile as the a priori between 18 - 125 km." Does that mean the the ACE retrieval of HCl uses the retrieved temperature from ACE measurements as input? This should be clarified.

**Author's response (2.)**
Yes, your understanding is correct. The ACE-FTS HCl retrieval procedure uses the temperature profile retrieved from the ACE-FTS measurements as input. We improved the manuscript as follows to clarify this point. We appreciate that you pointed it out.

**Author's changes in the manuscript (2.)**
-Page 20, Line 282, 283:
"The ACE-FTS retrieval procedure uses the retrieved temperature profile as the a priori between 18 - 125 km."
→ "The ACE-FTS HCl retrieval procedure uses the temperature profile retrieved from the ACE-FTS measurements between 18–125 km."

**Reviewer Comment (3.)**
Furthermore, in the explanation of the temperature retrieval for both instruments, it would help if it was mentioned what species or spectral features were actually used, in addition to specifying the a priori profiles.

**Author's response (3.)**

Thank you for your comment. The MLS temperature retrieval procedure used the $O_2$ lines at 118 and 239 GHz. The isotopic 239-GHz line is the primary source of temperature information in the troposphere, while the 118-GHz line is the primary source of temperature in the stratosphere and above [Liversey et al., 2018]. The ACE-FTS temperature profiles were retrieved from the $CO_2$ VMR profiles using hydrostatic equilibrium [Boone et al., 2020]. We added these descriptions to the manuscript.

**Author's changes in the manuscript (3.)**

-Page 20, Line 279, 280:

[revised manuscript text omitted]